

# uDALES 1.0.0: a large-eddy-simulation model for urban environments

Ivo Suter[1,2], Tom Grylls[1], Birgit Sützl[1], and Maarten van Reeuwijk[1]

[1]Imperial College London, London, UK
[2]Empa, Swiss Federal Laboratories for Materials Science and Technology, Dübendorf, Switzerland

**Correspondence:** M. van Reeuwijk (m.vanreeuwijk@imperial.ac.uk)

**Abstract.** Urban environments increasingly move to the fore of climate and air quality research due to their central role in the population's health and well-being. Tools to model the local environmental conditions, urban morphology and interaction with the atmospheric boundary layer play an important role for sustainable urban planning and policy-making. uDALES is a high-resolution, building-resolving large-eddy simulation code for urban microclimate and air quality. uDALES solves a surface energy balance for each urban facet and models multi-refection shortwave radiation, longwave radiation, heat storage and conductance, as well as turbulent latent and sensible heat fluxes. Vegetated surfaces and their effect on outdoor temperatures and energy demand can be studied. Furthermore a scheme to simulate emissions and transport of aerosols and some reactive gas species is present. The energy balance has been tested against idealized cases and the particle dispersion against field measurements, yielding satisfying results. uDALES can be used to study the effect of specific new constructions and building measures on the local micro-climate; or to gain new insight about the general effect of urban morphology on local climate, ventilation and dispersion. uDALES is available online under GNU General Public License and remains under maintenance and development.



# 1 Introduction

With an ever-increasing number of people living in cities (UNFPA, 2012), understanding how the urban morphology influences
the transport of momentum, heat, moisture and pollutants is central to designing healthy, sustainable and safe urban environments. Indeed, the large number of tall towers being erected in large cities around the world necessitate elaborate studies on how these affect the safety of pedestrians, since these structures channel high-momentum air down to ground level (Blocken et al., 2016). Densely populated areas are prone to develop an urban heat island (UHI) effect (increased temperatures relative to the surrounding rural areas partially due to the limited vegetation and water surfaces; Oke, 1982; Rosenzweig et al., 2015).
The UHI increases the intensity of heat waves, with a measurable increase in mortality rates (Pyrgou and Santamouris, 2018). Furthermore, the large concentrations of people in urban areas create problems with urban air quality; it is a sobering fact that health quality standards are exceeded in most of the world's largest cities, despite the known adverse effects on human health (World Heath Organization, 2016).

    The increased likelihood of extreme weather events due to climate change (IPCC, 2014) and the need to transition to a less
carbon-hungry society makes our ability to predict the urban climate even more important. Design decisions that are taken now will strongly influence whether we will be able to meet the intentions outlined in the Paris agreement (UNFCCC, 2016). The choice of building materials and city layout has to be reconsidered together with intelligent management of water systems and green spaces. Greening of façades and roofs can mitigate the effects of the UHI effect (Santamouris, 2014), specifically, it improves thermal comfort and lowers heat stress due to enhanced evapotranspiration from vegetation. Green roofs or walls also
lead to lower indoor temperatures and reduce the energy demand for air conditioning (Castleton et al., 2010). Consequently, the augmented installation of green infrastructure is a possible remedy for exceedingly hot cities and the subsequent health problems.

    With business-as-usual emission regulation policies, mortalities due to outdoor air quality will continue to rise in the period up to 2050 (Lelieveld et al., 2015). The 'metabolic' consumption of energy and materials within cities results in the presence
of a wide range of harmful pollutants in the urban atmosphere (Oke et al., 2017). Long-term improvements in air quality must be driven through reduced emissions however 1) this process requires large-scale, long-term infrastructural and behavioural change and 2) recent studies indicate the complexity and global variability of source contributions (e.g. the importance of non-combustible emission sources; Lelieveld et al., 2015; Grylls, 2020). Short-term and more flexible solutions are therefore also needed; e.g. temporary limitations on transport (Borge et al., 2018), optimised urban design (Llaguno-Munitxa and Bou-Zeid,
2018) and technical innovations for active removal (Sikkema et al., 2015). Accurate predictions of the emission, dispersion, chemical reaction and removal of pollutant species in the urban atmosphere is integral towards developing our understanding of and therefore solutions to the global challenge of urban air quality.

    Reliable methods to model the interaction between the atmospheric boundary layer and the urban morphology are key to predictions of pedestrian safety, urban microclimate and urban air quality. The urban structure is inherently heterogeneous:
roads, residential, commercial and industrial zones segment cities and are interspersed with vegetation and open water. Moreover, the urban fabric, i.e. the material composition and texture of the urban surface, is also subject to great variability. This



represents a major challenge to the modelling of urban processes due to their large range of associated spatial and temporal scales.

The complexity of the urban structure and fabric makes computational fluid dynamics (CFD) models a popular modelling

choice. Wind engineering models are predominantly based on the Reynolds-averaged Navier-Stokes (RANS) equations, which require turbulence parameterisations for the full range of active scales in the flow field Blocken (2015). This allows for the use of relatively large time-steps, or even steady-state simulations that predict the flow field reasonably well. However, the reliance on turbulence parameterisations requires careful validation and sensitivity analyses Blocken (2015). In non-neutral situations, which is the predominant atmospheric state over cities (Barlow, 2014; Kotthaus and Grimmond, 2018), the use of RANS is

more problematic due to the influence of buoyancy on the velocity field (Hanjalić and Kenjereš, 2008). Popular codes for RANS are any of the commercial CFD packages (Fluent, ANSYS CFX, COMSOL etc), although there are also open source alternatives (e.g. OpenFoam).

However, most CFD models neither contain a representation of energy exchanges at the urban surface nor explicitly treat radiative fluxes. Coupling of stand-alone energy balance models is a possibility (Musy et al., 2015), yet technical difficulties

persist and different approaches in model design exist across numerous research groups. The exceptions are urban climatology models based on the Reynolds-averaged Navier-Stokes (RANS) equation that are able to take into account the thermal exchanges in the urban area e.g. MIMO (Ehrhard et al., 2000), MITRAS (Schlünzen et al., 2003), ENVI-met (Huttner, 2012) and MUKLIMO3 (Sievers, 2016).

Large-Eddy simulation (LES) tools explicitly resolve the bulk of the turbulent scales in the flow, and are therefore less reliant

on turbulence models than RANS models. As they resolve the bulk of the energy-containing scales in the turbulence, they are more computationally demanding than RANS models. However, the continuous increase in computational resources makes LES models the most promising method to provide sufficiently detailed and accurate information on how the interaction of heat, moisture and momentum exchanges affect the local urban microclimate. Examples of LES models capable of dealing with the urban terrain are PALM-USM (Maronga et al., 2015; Resler et al., 2017) and OpenFoam2.

The aim of this paper is to present a new large-eddy simulation model for the urban environment, uDALES. This model is an extension of the atmospheric LES model DALES (Heus et al., 2010; Tomas et al., 2015). It can be used to model both the standard idealised flow cases that are pursued in air quality studies (Caton et al., 2003) and realistic case studies (Grylls et al., 2019). uDALES is capable of solving a surface energy balance in a two-way coupled manner with the LES solver and models the dominant processes that shape the urban environment, such as radiation, effect of vegetation, energy and momentum fluxes

from urban surfaces.

The paper is structured as follows. In section 2 the fundamentals of the model are outlined and the treatment of boundary conditions is detailed with a description of the treatment of the wall-fluid interaction. Furthermore the scheme to simulate emissions, transport and reactions of aerosols and reactive gas species is introduced. A new method for simulating the surface energy balances is introduced in section 3 together with the procedures for long- and shortwave radiation. Section 4 is used

to validate the new model against an urban measurement campaign for the dispersion of air pollution (DAPPLE), and an





urban energy balance model. Furthermore the simulation of vegetated surfaces is showcased. Finally in section 5 the future applications and development of uDALES are summarized.

## 2 Model description

### 2.1 Governing equations

The governing equations solved in uDALES (Heus et al., 2010) are within the Boussinesq approximation:

$$\frac{\partial \tilde{u}_i}{\partial x_i} = 0, \tag{1}$$

$$\frac{\partial \tilde{u}_i}{\partial t} = -\frac{\partial \tilde{u}_i \tilde{u}_j}{\partial x_j} - \frac{\partial \pi}{\partial x_i} + \frac{g}{\theta_0}\tilde{\theta}_v \delta_{i3} + F_i - \frac{\partial \tau_{ij}}{\partial x_j}, \tag{2}$$

$$\frac{\partial \tilde{\varphi}}{\partial t} = -\frac{\partial \tilde{u}_j \tilde{\varphi}}{\partial x_j} - \frac{\partial R_{\varphi,j}}{\partial x_j} + S_\varphi. \tag{3}$$

Here $u_i$ is the component of the velocity vector along the base vector $x_i$. Time is denoted as $t$. The modified pressure is $\pi = \frac{\tilde{p}}{\rho_0} + \frac{2}{3}e$, where $p$ is the pressure, $\rho$ is the density of air and $e$ is the subfilter-scale turbulence kinetic energy. Denoted by $\theta_v$ is the liquid water virtual potential temperature, $\delta_{ij}$ is the Kronecker delta, $F_i$ represents other forcings and $\tau$ is the deviatoric part of the subfilter momentum flux. The variable $\varphi$ can represent humidity, temperature, particulates or chemical species. The subfilter-scale scalar fluxes are denoted by $R_\varphi$ and the scalar sources or sinks by $S_\varphi$. Where multiple indices (i, j) appear,

summation following Einstein notation is implied. The tildes denote spatially filtered mean variables. Filtering in uDALES is implicit: the grid itself acts as the low-pass filter.

The buoyancy, which exerts a force in the vertical direction, is given by the liquid water virtual potential temperature approximated following Emanuel (1994):

$$\theta_v = \theta\left(1 - \left(1 - \frac{R_v}{R_d}\right)q_t - \frac{R_v}{R_d}q_l\right), \tag{4}$$

with

$$\theta = T\Pi^{-1}, \text{ and } \Pi = \left(\frac{p}{p_0}\right)^{\frac{R_d}{c_d}}, \tag{5}$$

where $\theta$ is the potential temperature, $\Pi$ is the Exner function, $c_d$ is the heat capacity of dry air, $R_d$ is the gas constant for dry air and $R_v$ is the gas constant for water vapour. This representation explicitly includes the effect of temperature, water vapour content and water droplets.

The subfilter stresses are modelled as

$$\tau_{ij} = -K_m\left(\frac{\partial \tilde{u}_i}{\partial x_j} + \frac{\partial \tilde{u}_j}{\partial x_i}\right), \qquad R_{\varphi,j} = -K_h\frac{\partial \tilde{\varphi}}{\partial x_j}, \tag{6}$$

where $K_m$ represents the eddy viscosity and $K_h$ the eddy diffusivity. The Vreman subgrid-scale model (Vreman, 2004) is used to obtain the eddy viscosity. This model is of similar complexity as the commonly used Smagorinsky model (Smagorinsky,





1963), but it behaves better in near-wall regions. The eddy diffusivity is related to the eddy viscosity via a constant turbulent
Prandtl number. uDALES is based on an Arakawa C-grid (Arakawa and Lamb, 1977). Pressure and scalar variables ($\varphi$) are
defined at cell centres and the three velocity components ($u, v, w$) are defined at the west, south and bottom sides of the grid
cells respectively. Numerical advection is calculated via a second-order central difference scheme for all prognostic fields
except for the pollutant concentrations where a $\kappa$-advection scheme (Hundsdorfer et al., 1995) is used to ensure monotonicity.
Time integration is done by a third-order Runge-Kutta scheme, following Wicker and Skamarock (2002). uDALES is written
in Fortran and the code is parallelised using MPI. For the parallelisation, the domain is sliced along the $y$-direction.

## 2.2 Immersed boundary method

The solid-fluid boundary between the air and the urban form is defined through the immersed boundary method (IBM) and wall
functions. The IBM allows for obstacles to be placed inside the fluid domain. It works on the principle that solid boundaries can
be modelled by adapting the body force terms in equations (2) and (3) in the cells that are adjacent to the defined boundaries
(Mittal and Iaccarino, 2005). This technique is in contrast to other methods where the computational grid is defined such that
it conforms to the solid-fluid boundary. The advantages of using the IBM are that the grid is relatively simple to generate and
that the pressure field can be solved directly via a fast Fourier-based transform method. The IBM introduced into uDALES
by Tomas et al. (2015) is defined such that the obstacles must conform to the defined Cartesian grid (Pourquie et al., 2009).
Boundaries are defined at the cell faces such that the normal component of the velocity can be set to zero (and therefore to
ensure that there is no advective flux through the wall by construction). The subgrid-scale flux terms acting across this plane
are also nullified. This version of the IBM is second-order accurate but limits the obstacles modelled in the domain to be
rectangular and grid-conforming.

## 2.3 Boundary conditions

### 2.3.1 Top, bottom and immersed boundaries

Various boundary conditions (BCs) can be selected for top, bottom, lateral and immersed boundaries. The BC for momentum at
the top can be given either as a fixed flux (Neumann type) or fixed value (Dirichlet type), where zero-flux (free slip) is the most
common setting. On the bottom and on the immersed boundaries, a no-slip condition for momentum is imposed via the wall
function discussed in section 2.2. Both Neumann and Dirichlet boundaries can also be applied for scalar quantities at the top of
the domain. For moisture and temperature, the immersed and bottom boundaries can either be prescribed as a flux or a boundary
value. In the case that the Dirichlet condition is chosen, the resulting temperature and moisture fluxes are obtained from the
wall function and optionally a surface energy balance can be solved to update the boundary values dynamically, as discussed
in detail in section 3. For other scalar quantities such as tracers and chemical species, the surface fluxes are prescribed.





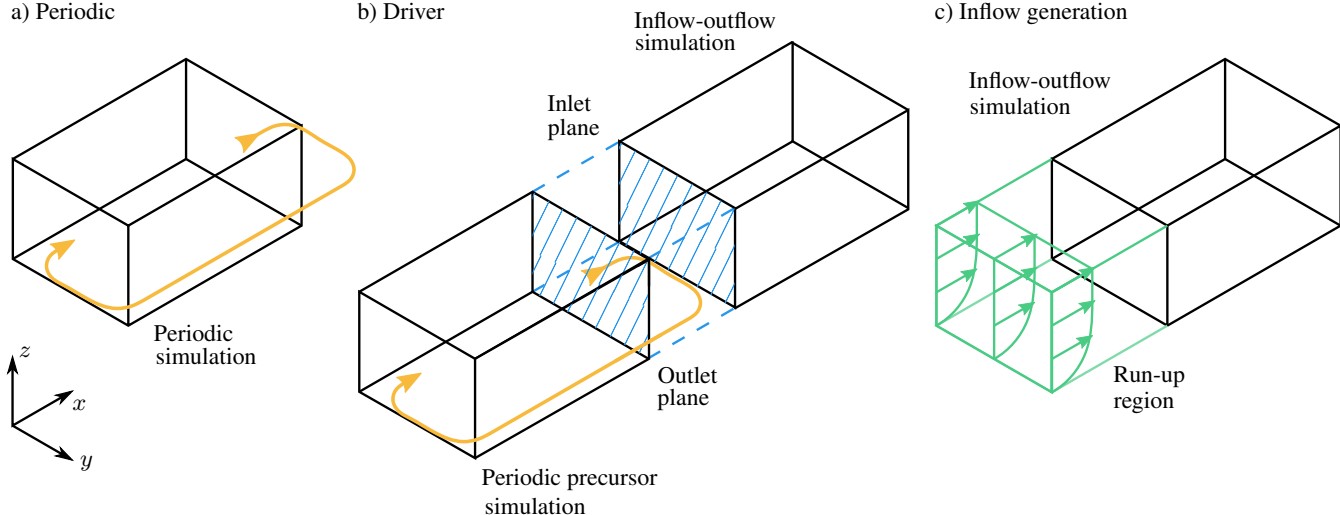

**Figure 1.** Diagrammatic view of the lateral boundary conditions available in uDALES. (a) Periodic simulation set-up, (b) driver simulation set-up (composed of a periodic precursor simulation and an inflow-outflow target simulation) and (c) inflow generation set-up (turbulent profiles generated numerically at the inlet and a run-up region in the streamwise direction to allow the flow to develop).

### 2.3.2 Lateral

The lateral boundary conditions in the $x$-direction can be set as periodic or 'inflow-outflow' (using either Neumann and Dirichlet conditions). In either case the $y$-direction remains periodic. The inflow condition can be defined by either

– running a precursor simulation from which outlet planes are used to 'drive' a later simulation (see figure 1b);

– following the cell perturbation method (see figure 1c; Kong et al., 2000; Tomas et al., 2015).

The three main simulation set-ups are shown diagrammatically in figure 1. The lateral boundary conditions are important in determining the planetary boundary layer flow that is modelled in uDALES.

### 2.4 Wall functions

Nearly all urban surfaces can be considered to be aerodynamically rough. This means that the roughness elements on a surface are much larger than the thickness of the viscous sublayer which is adjacent to every interface between a fluid and a smooth surface. Mean turbulent quantities can thus vary drastically close to walls and processes near the wall cannot be resolved on the grid and therefore need to be modelled. Processes close to vertical walls are qualitatively different from horizontal walls, particularly in the presence of buoyancy (e.g. Hölling and Herwig, 2005). However, vertical walls are commonly treated the same way as horizontal walls, namely the formation of a constant stress layer with logarithmic wind profile is assumed. This is due to the lack of established alternatives.





### 2.4.1 Wall-functions for momentum and temperature

For atmospheric flows wall functions on horizontal surfaces are commonly based on similarity laws determined by Buckingham
Π analysis. Such similarity laws for the surface layer have been known since the fundamental paper of Obukhov (1946) and
subsequent work together with A. Monin (Monin and Obukhov, 1954). A wall function for momentum and temperature based
on Uno et al. (1995) was introduced into uDALES, in which the eddy fluxes $u_*^2$ and $\theta_{v*} u_*$ are given by

$$u_*^2 = u^2 \frac{\kappa^2}{\ln{(z/z_0)}^2} F_{\mathrm{m}}(z, z_0, z_{0h}, Ri_B), \tag{7}$$

$$\theta_{v*} u_* = u \Delta \theta_{v} \frac{\kappa^2}{Pr \ln{(z/z_0)}^2} F_{\mathrm{h}}(z, z_0, z_{0h}, Ri_B), \tag{8}$$

where $z_0$ is the momentum roughness length, $z_{0h}$ is the roughness length for heat, $\kappa$ is the von Kármán constant and the func-
tions $F_{\mathrm{m}}$ and $F_{\mathrm{h}}$ describe the relationship between the atmospheric temperature and wind profile (Bulk Richardson number,
$Ri_B$) and the corresponding surface fluxes.

Due to the fact that surfaces are generally not dynamically smooth, momentum transfer has a strong dependence on the form
drag introduced by the pressure differences across those obstacles. However, for scalar transport no equivalent principle to pres-
sure exists. In fact, it has been shown (Garratt, 1994) that for rough surfaces scalar transport becomes less efficient compared
to momentum transport near the wall. We thus adopt the approach of Uno et al. (1995) and use a separate roughness length
$z_{0h} \leq z_0$ for scalars to approximate the parametric functions $F_{\mathrm{m}}$ and $F_{\mathrm{h}}$. This one-step iterative approach has been employed in
LES studies before (Cai, 2012a, b) and is very attractive given the fact that the computational costs are low. The same method
is used for vertical walls as well, where $z$ and $Ri_B$ are defined perpendicular to the wall. By doing this, the Richardson number
loses its context and one must be careful interpreting its values. For more details see Suter (2019).

To verify the correct implementation of the wall function, results from uDALES are compared directly to LES results from
Cai (2012a). The simulation set-up is a flow over a canyon-like cavity. The canyon is $h = 18\,\mathrm{m}$ high and $l = 18\,\mathrm{m}$ wide. The
domain size is $24\,\mathrm{m} \times 40\,\mathrm{m} \times 90\,\mathrm{m}$ with $80 \times 40 \times 91$ grid cells. This results in a resolution of $0.3\,\mathrm{m} \times 1\,\mathrm{m} \times 0.3\,\mathrm{m}$ inside the
canyon; and a grid that is stretched in the vertical direction above. The simulation is periodic in $x$ and $y$ (figure 1a) and driven
by a constant free-stream velocity $U_F = 2.5\,\mathrm{m\,s^{-1}}$. Two cases are examined: 1) the downstream wall and roof are heated ($T_0$+9
K, assisting case) and 2) the upstream wall and roof are heated ($T_0$+9 K, opposing case). In Case 1 the buoyancy flux from the
heated downstream wall will assist the recirculation forming inside the canyon, while in Case 2 the additional buoyancy will
oppose the general flow in the cavity. In Figure 2 the mean temperature fields for both cases are compared and show excellent
agreement. In the assisting case, the heat is concentrated on the downstream canyon edge from where it is efficiently mixed
with air aloft. The mean temperature inside the canyon is thus only slightly elevated. In the opposing case, the downdraft along
the heated wall leads to a more complicated flow patterns inside the canyon, effectively reducing the exchange between the
canyon and the atmosphere. As a result, more heat is trapped inside the canyon, leading to increased temperatures. The quanti-
tative agreement with Cai (2012a) confirms the correct implementation of the wall function for temperature for both horizontal
and vertical walls.





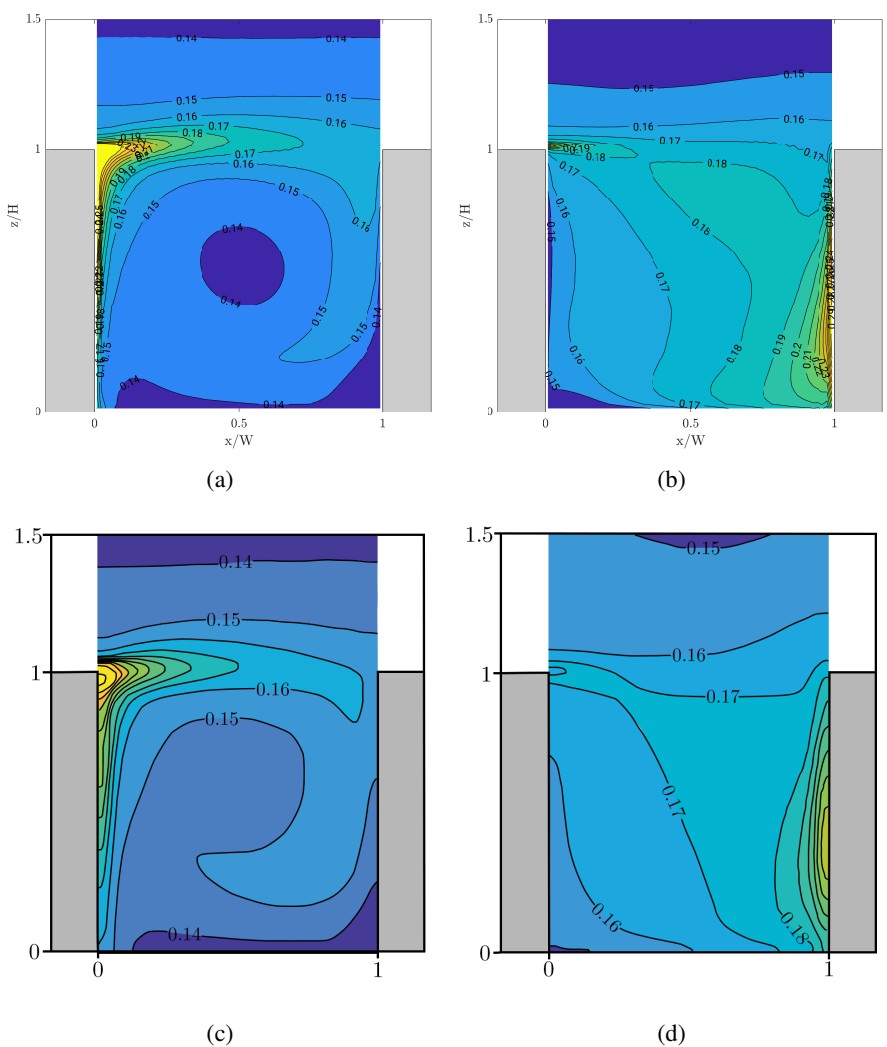

**Figure 2.** Mean temperature fields (a&b) compared to results reproduced from Cai (2012a, c&d). $T_0$ was subtracted. Left column shows the assisting case, right column the opposing case.

### 2.4.2 Wall function for moisture

The surface moisture fluxes require additional consideration. In uDALES it is assumed that water is stored in the soil and in vegetation, where any such surface has a water balance:

$$\frac{\mathrm{d}W}{\mathrm{d}t} = P - \frac{E_\star}{L_v}, \tag{9}$$

where $P$ [kg m$^{-2}$ s$^{-1}$] is the water supply rate, $E$ is the latent heat flux [W m$^{-2}$] and $L_v$ [J kg$^{-1}$] is the latent heat of vaporization. The behaviour of plants under various environmental conditions has been studied extensively (Moene and van Dam, 2014). However, of major interest are the exchanges of momentum, heat and water between the surface and the atmosphere.





Apart from water availability in the top soil, the evaporation from vegetation is determined mainly by the solar radiation reaching the leaves. The phase change from liquid water into water vapour in the air requires large amounts of energy, generally provided by the radiation. Furthermore, the pores in the leaves, stomata, are open during the daytime to allow the exchange of gases used in photosynthesis, facilitating transpiration. Tall vegetation like trees also provide shade for lower surfaces. While trees are indeed a very interesting topic in the research area of the urban climate, trees and tall vegetation are not considered here.

A wall function for moisture fluxes from vegetated surfaces was newly developed for uDALES. The moisture flux from roof and wall facets can be described identically to the heat flux (Eq. (8)). However, it is not obvious what the moisture difference $\Delta q$ between the wall and the atmosphere is, because water is usually not available directly on the surface and therefore one cannot assume that the air close to the surface is saturated with water vapour. Furthermore, open water surfaces are as of yet uncommon on buildings and most of the flux will thus stem from vegetation or soil of green roofs/walls. Plants lose water by opening their stomata, but also through their outermost layer, the cuticle. A large number of processes are involved in the transport of water in soil and plants. Indeed, the nature of the ground influences the roots and their water uptake; plants have varying metabolisms depending on age, size, season etc.. Modelling these processes is seldomly done in detail in atmospheric models. We thus utilise a modified Jarvis-Stewart (JS) approach (Jarvis, 1976; Stewart, 1988). This approach adds additional resistance to the transport of moisture from vegetation and soil to the atmosphere. The added resistance is based on empirical functions depending on plant and environmental parameters. The resistances for the plant canopy ($r_c$ [s m$^{-1}$]) and for the soil ($r_s$ [s m$^{-1}$]) can be expressed as:

$$r_c = \min\left(r_{\max}, \frac{r_{c,\min}}{D}f_1(K)f_2(W_G)f_3(T_s)\right), \tag{10}$$

$$r_s = \min\left(r_{\max}, r_{s,\min}f_2(W_G)\right), \tag{11}$$

where $D$ [m$^2$ m$^{-2}$] is the leaf area index, the term $f_1$ is a function of net shortwave radiation $K$ [W m$^{-2}$], $f_2$ depends on soil moisture content $W_G$ [kg m$^{-2}$] and $f_3$ on the wall temperature $T_s$ [K]. Following Van den Hurk et al. (2000) for $f_1$ and $f_2$ and Moene and van Dam (2014) for $f_3$, these are given by:

$$f_1^{-1} = \min\left(1, \frac{c_1 K + 0.05}{0.81(c_1 K + 1)}\right), \tag{12}$$

$$f_2^{-1} = \min\left(1, \max\left(0.001, \frac{W_G/d - W_{\text{wilt}}}{W_{\text{fc}} - W_{\text{wilt}}}\right)\right), \tag{13}$$

$$f_3^{-1} = \max\left(0.001, (1 - c_2(c_3 - T_s)^2)\right), \tag{14}$$

$$h_{\text{rel}} = \max\left(0, \min\left(1, \frac{1}{2}\left[1 - \cos\left(\pi\frac{W_G}{W_{\text{fc}}}\right)\right]\right)\right), \tag{15}$$

where $c_1 = 0.004$ m$^2$ W$^{-1}$, $c_2 = 0.0016$ K$^{-2}$, $c_3 = 298$ K, $d$ [m] is the thickness of the soil layer, $W_{\text{wilt}}$ [kg m$^{-3}$] is the soil moisture at the wilting point, $W_{\text{fc}}$ [kg m$^{-3}$] is the soil moisture at field capacity and $h_{\text{rel}}$ [-] is the relative humidity above soil following Noilhan and Planton (1989). The moisture flux from roof and wall facets is then expressed as

$$q_* u_* = \frac{q_a - q_{\text{sat}}(T_s)}{\frac{1}{uC_h} + r_c} + \frac{q_a - q_{\text{sat}}(T_s)h_{\text{rel}}}{\frac{1}{uC_h} + r_s}, \tag{16}$$





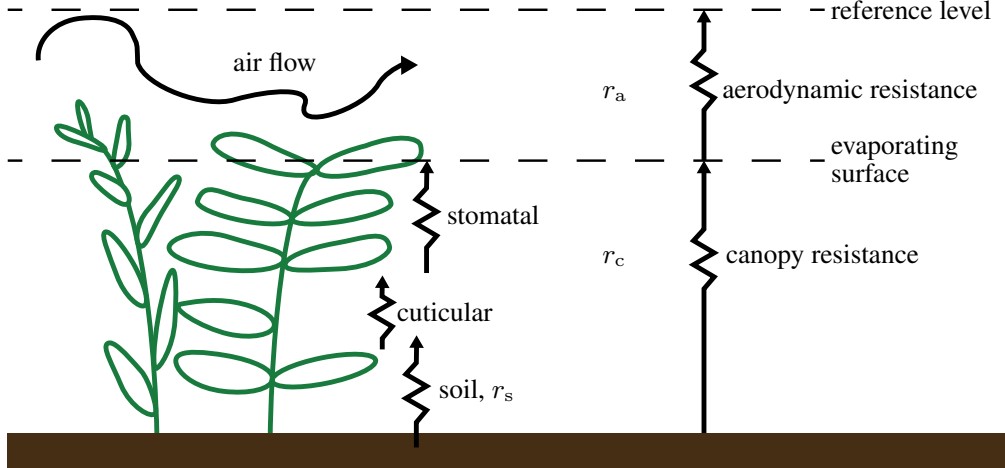

**Figure 3.** Diagram of the resistances encountered in a simplified soil-plant-atmosphere system.

with the units [kg kg$^{-1}$ m s$^{-1}$]. The heat transfer coefficient $C_\mathrm{h} = \frac{\kappa^2}{Pr\ln(z/z_{0h})^2} F_\mathrm{h}(z, z_0, z_{0h}, Ri_B)$ is obtained from the standard wall function (Eq. (8)) where the vegetation is considered with appropriate roughness lengths. To determine the saturation

specific humidity $q_\mathrm{sat}$ [kg kg$^{-1}$] we use empiric formulas from Bolton (1980) and Murphy and Koop (2005) that are accurate within normal temperature ranges encountered in cities.

## 2.5 Emissions and chemistry

A key application of urban LES models is the study of urban air pollution. This entails modelling the life cycle (emission, dispersion, chemical reaction and removal) of pollutants within the LES domain at the relevant (microclimate) scales. The high-

resolution, time-resolving nature of uDALES makes it able to accurately resolve the transport of passive scalar fields within the turbulent urban flow field. The capability to model both idealised (point and line sources with initial Gaussian distributions) and realistic emissions (via networks of volumetric point sources) has been introduced into uDALES. The chemistry of the null cycle has also been added to capture the reactions between nitrogen oxides ($NO_x$) and ozone ($O_3$; Grylls et al., 2019).

Idealised emission sources are integral for fundamental studies of urban pollution dispersion (e.g. line sources within infinite

canyons; Caton et al., 2003). Point sources are used to represent passive scalar releases in the validation of uDALES in section 4.1. Realistic traffic emissions are obtained by coupling the LES model with a traffic microsimulation and emissions model within the preprocessing routine. By rasterising the resulting pollutant emissions to the grid used in uDALES, traffic emissions can be read into the LES model and simulated via a network of pollutant sources at the lowest level of the domain. Grylls et al. (2019) used the traffic microsimulation model VISSIM (PTV AG, 2017) and an instantaneous, general regression emissions

model (Luc Int Panis, 2006) to use uDALES to study pollution dispersion over a case study region in London, UK.

Chemical reactions occurring at a similar timescale to the dispersion of pollutants within the urban canopy layer (UCL) significantly alter the spatial and temporal evolution of the affected pollutant species. This phenomena is particularly important





in accurately capturing Nitrogen Dioxide, $NO_2$, and Ozone, $O_3$, concentrations. By time-resolving the relevant reactions within urban LES, a greater understanding can be gained of these harmful air pollutants and on the effect of these reactions on e.g.

pedestrian-level exposure. The chemistry of the null cycle, which has no net chemical effect (meaning that $NO_x$ and $O_x$ are conserved), follows

$$NO_2 + h\nu \quad \rightarrow \quad NO + O^\bullet, \tag{R1}$$

$$O^\bullet + O_2 \quad \rightarrow \quad O_3, \tag{R2}$$

$$NO + O_3 \quad \rightarrow \quad NO_2 + O_2. \tag{R3}$$

The photodissociation R1 has the rate constant $J_{\mathrm{NO_2}}$, and reactions R2 and R3 the first order rate constants $k_2$ and $k_3$, respectively. The chemical processes of the null cycle have been added into uDALES using reactions rates from Wallace and Hobbs (2006). The relatively high reactivity of the Oxygen radical, $O^\bullet$, permits a simplification of the null cycle leading to the following prognostic relationships (Zhong et al., 2017):

$$\epsilon_{\mathrm{NO}} \quad = \quad J_{\mathrm{NO_2}}\,[\mathrm{NO_2}] - k_3\,[\mathrm{NO}]\,[\mathrm{O_3}], \tag{17}$$

$$\epsilon_{\mathrm{NO_2}} \quad = \quad -J_{\mathrm{NO_2}}\,[\mathrm{NO_2}] + k_3\,[\mathrm{NO}]\,[\mathrm{O_3}], \tag{18}$$

$$\epsilon_{\mathrm{O_3}} \quad = \quad J_{\mathrm{NO_2}}\,[\mathrm{NO_2}] - k_3\,[\mathrm{NO}]\,[\mathrm{O_3}], \tag{19}$$

where $\epsilon$ is an additional source term on the right-hand-side of equation 3 for reactive pollutant fields. The chemistry is implemented following a split fully implicit time-integration scheme. A validation of the chemical scheme can be found in Grylls et al. (2019).

# 3   Surface energy balance

Any attempt to understand the urban microclimate must start with an analysis of its surface energy balances (SEB; Oke et al., 2017). Urban areas exchange energy in various forms (Erell et al., 2011), such as incoming and outgoing longwave ($L^\downarrow$, $L^\uparrow$) and shortwave radiative fluxes ($K^\downarrow$, $K^\uparrow$), the turbulent sensible heat flux ($H$), the turbulent latent heat flux ($E$), ground heat flux ($G$) and the heat storage per unit time ($\mathrm{d}Q/\mathrm{d}t$, often denoted $\Delta Q$). These fluxes have to balance and the SEB is expressed

as

$$\frac{\mathrm{d}Q}{\mathrm{d}t} = (L^\downarrow - L^\uparrow) + (K^\downarrow - K^\uparrow) - (H + E + G) + O, \tag{20}$$

where all the terms have unit W m$^{-2}$. Several possible approaches for studying urban climate have been used, ranging from observation to numerical simulations. In this context, the physical processes determining the SEB and their complex interactions must be taken into account (Mirzaei and Haghighat, 2010; Mirzaei, 2015; Moonen et al., 2012; Arnfield, 2003; Rizwan

et al., 2008; Grimmond, 2007; Barlow, 2014).

A common approach is the use of standalone urban energy balance (UEB) models. UEBs are based on the concept that total energy in the urban boundary layer, the lowest few hundred meters of the atmosphere, has to be conserved. UEBs are





widely used as surface schemes in numerical weather prediction and regional atmospheric models, such as the Town Energy Budget (Masson, 2000, TEB) used by the French national weather service (Météo-France) or the Joint UK Land Environment

Simulator (Best et al., 2011; Clark et al., 2011, JULES) by the UK Met Office. They are computationally inexpensive and include many physical processes, but are still very simplified representations of the urban environment. The building geometry is often only represented in an averaged sense, and no single building can be studied. Furthermore, the transport in the air is parameterised. Other UEBs include variations of TUF (Krayenhoff and Voogt, 2007; Yaghoobian and Kleissl, 2012), SUEWS (Järvi et al., 2011), RayMan (Matzarakis et al., 2010) and the SOLWEIG model (Lindberg et al., 2008), Grimmond et al. (2010)

provides an overview and comparison of UEB models. We introduced the concept of a surface energy balance into uDALES, where every bottom and immersed boundary has its own material properties and all the energy fluxes from (20) and (9) are modelled. The surface properties $W_G$ and $T_s$ can vary in time and are spatially heterogeneous. Often Eq. (20) is considered for a large area, averaging out many local features: e.g. in urban energy balance models the horizontal scales are usually $10^2 - 10^4$ m (Grimmond et al., 2010). The time-resolutions is correspondingly coarse. The much higher resolution of uDALES allows

to consider individual surfaces, where all the terms of Eq. (20) become rather intricate. The turbulent sensible and latent heat fluxes change with every wind gust while the internal wall temperatures are much less sensitive and vary on much larger time scales. Furthermore, the radiation terms depend strongly on the orientation of the surface, shading and the field of view. For example, if a large portion of the field of view of a surface comprises of other surfaces it likely receives only little direct sunlight, since it is often shaded. However, it is likely to receive a large amount of longwave radiation from its surroundings

as building walls are generally warmer than the sky. Due to the large spatial and temporal variability in surface temperatures, especially in heterogeneous terrain, it is important to consider the surface energy balance carefully to achieve more realistic urban representation in LES models. To account for these effects, every surface in uDALES is divided into several smaller facets. The energy budget (20), including radiation, has to be evaluated for each facet.

## 3.1 Urban facets


uDALES accepts grid-conforming immersed boundaries (see section 2.2), i.e. cuboids, and for the calculation of the surface energy balance it is necessary that each side of those blocks (i.e. each facet) is either entirely internal or external. Accepting these restraints simultaneously leads to a finer representation of the surface. Most topographic elements, like buildings, are sliced into smaller blocks. An example is shown in Figure 4. Suspended and overhanging structures are currently not sup-

ported.

Facets represent all types of surfaces present in the urban landscape, such as roofs, walls, roads and green roofs. Every facet is thus also assigned properties accordingly, needed for the wall function and energy balance calculation. These are the momentum roughness length $z_0$ [m], the heat roughness length $z_{0h}$ [m], the shortwave albedo $\alpha$ [-] and the longwave emissivity $\epsilon$ [-]. Furthermore, every facet consists of multiple layers each with properties: thickness $d$ [m], density $\rho$ [kg m$^{-3}$], specific heat

capacity $c_p$ [J kg$^{-1}$ K$^{-1}$] and thermal conductivity $\lambda$ [W m$^{-1}$ K$^{-1}$]. The last three terms can be combined to form the thermal diffusivity $\varkappa = \lambda/(\rho c_p)$ [m$^2$ s$^{-1}$].



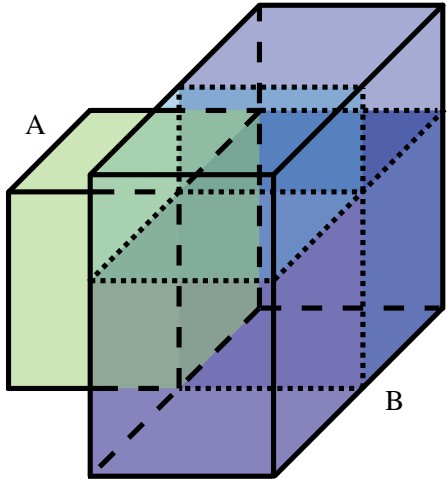

**Figure 4.** A building consisting of two blocks A and B. Block B will be divided into four smaller blocks, so all faces are either entirely on the in- or outside of the building.

Additional facets have to be introduced to populate the voids between building blocks (i.e. roads, parks, etc.). The floor facets immediately adjoining a building are given the same length as the corresponding wall facets and a width of one grid cell. The rest of the floor is then filled with rectangles, restrained by a maximum side length. Alternatively, if the nature and configuration of the floor is known in more detail the facets can be created accordingly and be specified using the facet properties.

Following Krayenhoff and Voogt (2007) we add a wall along the domain edge of a height equal to the median building height in the domain. This wall is used to approximate the effect of the surrounding built environment on radiation, that cannot be captured within the model domain. These bounding wall facets are solely used for radiative calculations and do not appear in the LES simulation and have thus no direct effect on air flow. An example of urban facets, including roads and bounding walls, can be seen in Figure 11a.

## 3.2 View factors

To calculate the radiative exchanges between two facets, their geometric relation to one another has to be determined and the radiative processes need to be described. To achieve this, there are commonly two alternative approaches pursued in complex urban topography: 1) ray tracing and Monte Carlo simulation; and 2) radiosity approach with configuration factors. For very complex or curved surfaces radiative exchange is commonly determined by tracing randomized rays of a Monte Carlo simulation with considerable computational effort. This approach becomes more and more feasible with increasing computational power and specialised GPU computing platforms. A big advantage of this approach is its potential to also handle specular reflections. Ray tracing models have for example been used to study urban photovoltaic energy potential (Erdélyi et al., 2014)



and urban climate and meteorology (Krayenhoff et al., 2014; Girard et al., 2017).

However, for planar surfaces, as are present in this study, the radiosity approach is generally faster (Walker et al., 2010). In this approach it is assumed that the radiosity across a facet is uniform. This allows the separation of the geometric relation between two facets from the radiative process itself (Howell et al., 2010). View factors (configuration factors) describe this

geometric relation. The radiosity approach has been used to study the urban climate (e.g. Aoyagi and Takahashi, 2012; Resler et al., 2017). The approach currently employed in uDALES follows Rao and Sastri (1996). It has to be noted at this point that several conventional assumptions were made regarding the radiation and radiative properties of all of the facets (e.g. Krayenhoff et al., 2014; Howell et al., 2010), namely: 1) no wavelength dependency, except the separation into short- and longwave; 2) facets are "grey" in the longwave regime, i.e. absorptivity = emissivity; 3) facets are isothermal; 4) reflections are diffuse; 5)

emitted radiation is diffuse; and 6) the radiosity (leaving radiant flux) is uniform across the facet.

For calculating both longwave and shortwave radiation, view factors are used. The view factor $\psi_{i,j}$ is defined as the ratio of radiation leaving the surface of facet $i$ hitting surface of facet $j$, divided by the total amount of radiation leaving facet $i$. View factors thus take values between 0 and 1 and $\sum_{j \in q} \psi_{i,j} = 1$, where $q$ is the set of all facets that can be seen by $i$ (including the sky). All view factors $\psi_{i,j}$ between any two facets $i$ and $j$ have to be calculated. This makes the number of calculations needed

$\mathcal{O}(n^2)$.

Several numerical methods exist to calculate view factors between surfaces, since in most cases the analytical solution to the problem is not known. The fact that view factors only depend on the geometry allows one to calculate all $\psi_{i,j}$ *a priori* and store the resulting matrix as an input to the LES model. Numerical integration was used to obtain the view factors in this study. Considerable computational savings can be achieved by replacing the integration over two areas by integration over two

surface boundaries. The view factor can then be expressed as (Siegel and Howell, 2001)

$$\psi_{i,j} = \frac{1}{2\pi A_i} \oint_{C_i} \oint_{C_j} \left[ \ln(S) \mathrm{d}x_j \mathrm{d}x_i + \ln(S) \mathrm{d}y_j \mathrm{d}y_i + \ln(S) \mathrm{d}z_j \mathrm{d}z_i \right], \tag{21}$$

where $S$ is the distance between two infinitesimal line segments along the contours of the facets ($C_i, C_j$). The numerical integration of $\ln(S)$ is done using $6^{\text{th}}$ order Gauss-Legendre quadrature, which is effective and accurate for straight-edged contours (Rao and Sastri, 1996). The evaluation of equation (21) is impossible in the case that the two contours share a

common edge, because $S = 0$. Ambirajan and Venkateshan (1993) provide an exact analytical solution for the contribution from the element on the common edge of the two surfaces to the overall view factor:

$$\Delta\psi_{i,j} = \frac{L_{\mathrm{c}}^2}{2\pi} \left( \frac{3}{2} - \ln(L_{\mathrm{c}}) \right), \tag{22}$$

where $L_{\mathrm{c}}$ is the length of the common edge. Since all facets are straight-edged, the only error in this approach lies in the approximation of $\ln(S)$ within the elemental intervals of the integration. High order quadratures thus ensure accurate numerical

results.

View factors can be calculated using equations (21) and (22) as long as both facets can see each other entirely. Three important





exceptions can be identified: 1) The facets cannot see each other at all, given their position and orientation; 2) a facet intersects the plane defined by the other facet; 3) the view is (partly) blocked by another surface. The first case occurs frequently. E.g. any west facing facet cannot possibly see any other west facing facet or any facet that is entirely to the east. This relationship

is reciprocal. The second problem can be circumvented by cropping the intersecting facet along the intersection line. The view factors can then be calculated using the now smaller facet and the result remains accurate. There is no straightforward solution to the third problem, assessing if the view between two facets is blocked. Depending on how the facets are arranged and the geometry of the object blocking their view, determining a precise view factor can be difficult. To avoid computationally expensive solutions such as Monte Carlo simulations we determine a percentage $p_\mathrm{b}$ that the facets can see of each other and

multiply $\psi$ with that percentage. To obtain $p_\mathrm{b}$ we define five points, the four corners and the centre, on every facet and determine if they can see the corresponding point on the other facet. The weight of the centre is $50\%$ and the weight of each corner $12.5\%$.

The sky view factor ($\psi_{\mathrm{sky},i}$) denotes the fraction of radiation leaving $i$ that enters the sky vault and does not impinge on any other facet. It is the residual view factor after summing over all facets: $\psi_{\mathrm{sky},i} = 1 - \sum_{\mathrm{j}=1}^{p} \psi_{i,j}$, where $p$ is the total number of facets. The sky view factor is also used to determine how much diffusive radiation from the sky any facet receives. For the

validation of the view factor calculations see Suter (2019).

### 3.3 Shortwave radiation budget

Assuming no transmission of radiation through the surface, the net shortwave radiation $K$ (W m$^{-2}$) on a facet $i$ is defined as

$$K_i = K_i^{\downarrow} - K_i^{\uparrow},\tag{23}$$

where the reflected shortwave radiation ($K^{\uparrow}$) can be expressed as

$$K_i^{\uparrow} = \alpha_i K_i^{\downarrow}.\tag{24}$$

The incoming shortwave radiation ($K_i^{\downarrow}$) can be divided into three parts (Oke et al., 2017): direct solar radiation ($S_i$), diffuse radiation from the sky ($D_i$) and diffusely reflected radiation from other facets ($R_i$):

$$K_i^{\downarrow} = S_i + D_i + R_i.\tag{25}$$

It is important to consider all three components. Many of the urban facets can be completely shaded, thus $S_i = 0$, but potentially

receive diffuse radiation from the sky or reflected radiation from other facets.

The direct solar radiation on a facet is defined as (Wu, 1995)

$$S_i = I \cos(\upsilon - \varphi_i) \cos(\chi_i) f_{\mathrm{e},i}.\tag{26}$$

The solar irradiance ($I$) is a model input and can be a constant or follow a diurnal cycle; it is assumed that all effects of atmospheric turbidity and clouds are included. The effect of geometry is captured by the sunlit fraction of the facet $f_{\mathrm{e},i}$, the

two cosines account for the orientation of the facet in relation to the sun. The angle $\upsilon$ is the solar zenith and $\varphi_i$ the slope angle





of the facet (i.e. $\varphi_i = 0$ for a horizontal facet and $\varphi_i = \pi/2$ for a vertical facet). The angle $\chi_i$ is constructed in the following way

$$\chi_i = \begin{cases} \pi/2, & \text{if surface faces away from sun} \\ 0, & \text{elseif the surface is horizontal} \\ |\Omega_h - \Omega_i|. & \text{else} \end{cases} \tag{27}$$

Where $\Omega_h$ is the solar azimuth and $\Omega_i$ is the azimuth of facet $i$. Like the solar irradiance, the solar zenith and azimuth angles
are also model inputs. Solar angles for all coordinates, dates and times can be readily obtained from online sources such as the "Solar Position Calculator" from the National Oceanic and Atmospheric Administration (NOAA, 2018).

The facet azimuth angles are also used to determine whether the facet is shaded or not. If $|\Omega_h - \Omega_i|$ is larger than $\pi/2$, the facet is not oriented towards the sun and is thus self-shaded. If the facet $i$ is not self-shaded the sun might still be blocked by
another facet $j$. Therefore, the path between every non-self-shaded facet and the sun is calculated and checked if it intersects with any other facet. The same principle as for view factors is used and the sunlit status is calculated for the four corners and the centre of the facet, where the centre contributes 50% and each corner 12.5% to the total sunlit fraction $f_{e,i}$.

The diffuse radiation ($D_i$) is caused by light scattering in the sky and is given by

$$D_i = \psi_{sky,i} D_{sky}, \tag{28}$$

where $D_{sky}$, the total diffuse sky radiation, is a model input. The sky view factor $\psi_{sky,i}$, is dependent on the urban geometry and has to be calculated for every facet. In equation (28) we ignore the directionality of $D_{sky}$, which is in reality anisotropic (Morris, 1969), and can thus use $\psi_{sky,i}$ directly to calculate the fraction of diffuse sky radiation impinging on $i$.

Shortwave radiation reflected by the environment has to be considered as well. All reflections are currently assumed to be perfectly diffusive (Lambertian), i.e. the amount of reflected light on a given surface is distributed equally in all directions. This is necessary to utilise view factors but does, by definition, not allow specular reflections. The amount of reflected light arriving at facet $i$ is thus given by

$$R_i = \sum_{j=1}^{p} \psi_{j,i} K_j^{\uparrow}, \tag{29}$$

where $\psi_{j,i}$ is the view factor between facet $j$ and $i$ and $p$ is the total number of facets.

The calculation of the short wave contribution is implemented into uDALES through a iteration algorithm. First view factors for all facet pairs $(i, j)$ are calculated and stored. The direct solar and diffuse sky radiation give an initial approximation for the shortwave budget of every facet. To account for multiple reflections, the calculation of $K_i^{\uparrow}$ (Eq. (24)-(29)) has to be iterated for
all facets using the following scheme:





- first reflection ($n = 0$)

    1. calculate initial $K_{i,0}^{\downarrow} = S_i + D_i$ using (26), (28)

    2. calculate initial $K_{i,0}^{\uparrow}$ using (24)

    3. calculate $R_{i,0}$ using (29)

    4. recalculate $K_{i,1}^{\downarrow} = K_{i,0}^{\downarrow} + R_{i,0}$

- iterate following steps for $n = 1, 2, ...$, until all changes are below a desired threshold $\epsilon_{\mathrm{k}}$:

    1. only the additional reflected radiation from the previous iteration is considered for further reflections, i.e. $K_{i,n}^{\uparrow} = \alpha_i R_{i,n-1}$

    2. recalculate $R_{i,n}$ based on the updated $K_n^{\uparrow}$

    3. update $K_{i,n+1}^{\downarrow} = K_{i,n}^{\downarrow} + R_{i,n}$

    4. calculate convergence criterion: $\max\limits_{i}\left((K_{i,n+1}^{\downarrow} - K_{i,n}^{\downarrow})/K_{i,n}^{\downarrow}\right) \leq \epsilon_{\mathrm{k}}$

Once the algorithm has converged, $K^{\downarrow}$ and also $K^{\uparrow}$ (via (24)) are known for all facets. This algorithm converges quickly, usually within 10 iterations for <1% error, since albedos are generally low resulting in a large fraction of the radiation being absorbed and additional parts being lost towards the sky with every cycle.

## 3.4 Longwave radiation

The net longwave radiation $L$ consists of four components.

$$L_i = \varsigma_{\mathrm{L},i}(\psi_{\mathrm{sky},i}L_{\mathrm{sky}}^{\downarrow} + L_{\mathrm{env},i}^{\downarrow} + L_{\mathrm{R},i}) - L_i^{\uparrow} \qquad [\mathrm{Wm}^{-2}], \tag{30}$$

where $\varsigma_{\mathrm{L},i}$ is the longwave absorptivity and $L_{\mathrm{env},i}^{\downarrow}$ is the incoming longwave radiation from other facets. Since $\varsigma_{\mathrm{L},i}$ is generally close to unity for normal building materials, no reflections are considered for longwave radiation and the incoming reflected radiation $L_{\mathrm{R},i} = 0$. The incoming longwave radiation from the sky ($L_{\mathrm{sky}}^{\downarrow}$) is given as a model input. $L_i^{\uparrow}$ is the outgoing longwave radiation depending on the surface temperature $T_{\mathrm{s},i}$ according to the Stefan-Boltzmann law $L_i^{\uparrow} = \sigma \epsilon_i T_{\mathrm{s},i}^4$ with the Stefan-Boltzmann constant $\sigma \approx 5.67 \times 10^{-8}$ W m$^{-2}$ K$^{-4}$ and $\epsilon_i$ is the emissivity of facet $i$. The incoming longwave radiation from the other facets is given by:

$$L_{\mathrm{env},i}^{\downarrow} = \sigma \sum_{\mathrm{j}=1}^{p} \psi_{j,i} \epsilon_j T_{\mathrm{s},j}^4 \qquad [\mathrm{Wm}^{-2}], \tag{31}$$

where $p$ again is the total number of facets, $\psi_{j,i}$ is the view factor between facet $j$ and $i$, $\epsilon_j$ is the emissivity of facet $j$ and $T_{\mathrm{s},j}$ is the surface temperature of facet $j$. The calculation of the longwave contribution does thus not require explicit iteration.





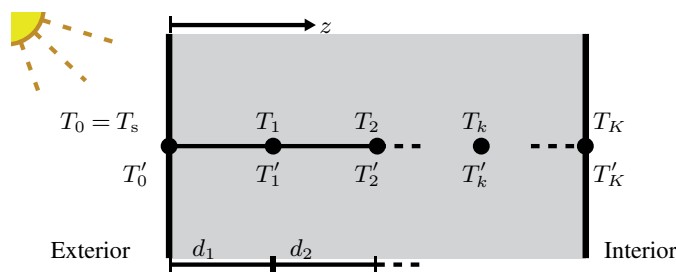

**Figure 5.** Grid layout used to model conductive heat flux.

### 3.5 Surface energy budget of ground, wall and roof facets.

To calculate the temperature evolution of every facet, we assume that every wall and roof consists of $K$ layers of building material (see Fig. 5). The temperature at the interface of layer $k \in \{1, ..., K-1\}$ and $k+1$ is defined as $T_k$. Furthermore, let $T_K$ be the indoor temperature of the facet and $T_s = T_0 = T(z = 0)$ be the outdoor surface temperature used for calculating radiation and convective heat fluxes. At every instant $T_0$ is given by the balance of energy fluxes at the surface:

$$-G_i(T_0) = K_i + L_i(T_0) + H_i(T_0)). \tag{32}$$

The net radiation on the right-hand side of the equation above is given by the radiation calculations discussed in the previous section. The values of $H$ are the time mean fluxes provided by the wall function, $K$ only depends on the position of the sun and the incoming longwave radiation $\varsigma_{\mathrm{L},i}(\psi_{\mathrm{sky},i} L^{\downarrow}_{\mathrm{sky}} + L^{\downarrow}_{\mathrm{env},i})$ can be calculated from the other facets temperatures. The two remaining unknown terms are $L_i^{\uparrow}$ and the ground heat flux $G_i$.

A defining aspect of this problem is that the boundary condition at the surface, which drives the energy exchange with the wall, is strongly non-linear and, under the assumption that the ground heat flux ($G_i$) is purely conductive, i.e. $G_i = \lambda \frac{\partial T}{\partial z}\big|_0$, which can be written as:

$$\lambda \left.\frac{\partial T}{\partial z}\right|_0 = \epsilon \sigma T_0^4 + r, \qquad \text{where } r = -(K + L^{\downarrow} + H), \tag{33}$$

$\lambda$ is the thermal conductivity in W m$^{-1}$ K$^{-1}$ and $z$ is pointing into the layer. This boundary condition requires information about both the temperature and its gradient at the boundary, and it would thus be very useful to have a numerical method for which these both are available at the same location without approximation. To achieve this, the variables are distributed in pairs of the temperature $T_k$ and its gradient $T' = \frac{\partial T_k}{\partial z}|_k$ on the cell edges, implying that there are $2(K+1)$ unknowns if the wall is discretised into $K$ layers. This grid layout is not conventional, but is closely related to Hermitian interpolation (Peyret and Taylor, 2012) and has been used successfully to predict non-hydrostatic free-surface flow (Van Reeuwijk, 2002).

A solution will be constructed using a set of piecewise continuous quadratic polynomials, which will turn out to be splines. Consider a second order polynomial

$$T(z) = a + bz + cz^2, \tag{34}$$





valid over the interval $z_k < z < z_{k+1}$. Defining $T(z_k) = T_k, T(z_{k+1}) = T_{k+1}, T'(z_k) = T'_k$ and $T'(z_{k+1}) = T'_{k+1}$, it is straight-forward to demonstrate that the quadratic is consistent with the following relation between the temperatures and its derivatives

$$\frac{1}{2}\left(T'_k + T'_{k+1}\right) = \frac{T_{k+1} - T_k}{d_{k+1}},\tag{35}$$

where $d_{k+1} = z_{k+1} - z_k$. The coefficients $a, b, c$ are determined by requiring that $T(z_k) = T_k, T(z_{k+1}) = T_{k+1}$ and $T'(z_k) =$

$T'_k$, with result

$$T(z) = \frac{(z - z_k)(z_{k+1} - z)}{2d_{k+1}}\left(T'_k - T'_{k+1}\right) + \frac{z_{k+1} - z}{d_{k+1}}T_k + \frac{z - z_k}{d_{k+1}}T_{k+1}.\tag{36}$$

Here, (35) was used to make the representation symmetrical in the arguments. By evaluating the temperature and its gradient at $z_k$ and $z_{k+1}$ it becomes clear that (36) indeed satisfies the requirements of a spline, as a set of these functions produces a curve which is continuous in both the temperature and the gradient of the temperature. An important property of (36) is that

the mean temperature in the layer is given by

$$\frac{1}{d_{k+1}}\int_{z_k}^{z_{k+1}} T(z)dz = \frac{1}{2}(T_k + T_{k+1}) + \frac{d_{k+1}}{12}(T'_k - T'_{k+1}).\tag{37}$$

The temperature evolution inside the wall is described by an unsteady heat equation of the form

$$\frac{\partial T}{\partial t} = \frac{\partial}{\partial z}\left(\varkappa\frac{\partial T}{\partial z}\right),\tag{38}$$

where $\varkappa$ is the thermal diffusivity in m$^2$ s$^{-1}$. Integrating this equation over the interval $z_k < z < z_{k+1}$ and substituting (37)

results in the relation

$$\frac{\mathrm{d}}{\mathrm{d}t}\left(\frac{d_{k+1}}{2}(T_k + T_{k+1}) + \frac{d_{k+1}^2}{12}(T'_k - T'_{k+1})\right) = \varkappa(z_{k+1})T'_{k+1} - \varkappa(z_k)T'_k.\tag{39}$$

Equations (35) and (39) provide $2K$ equations. The final two equations are provided by the boundary conditions. The boundary condition at $z = 0$ is the right-hand side of the surface energy balance equation 33, where the outgoing longwave radiation is expressed in the form $L_i^{\uparrow} = \left(\epsilon_i\sigma T_0^3\right)T_0$. At the building interior a constant temperature $T_K = T_B, \mathrm{d}T_K/\mathrm{d}t = 0$ is imposed.

In matrix form, the system of equations for $K$ layers is given by

$$A\mathbf{T}' = \mathbf{b} + B\mathbf{T},\tag{40}$$

$$C\frac{\mathrm{d}}{\mathrm{d}t}\mathbf{T} + D\frac{\mathrm{d}}{\mathrm{d}t}\mathbf{T}' = E\mathbf{T}'.\tag{41}$$

The matrix A is determined by the left hand side and and matrix B by the right hand side of Eq. (35). The vector $\mathbf{b}$ contains the sum of the surface energy fluxes (without the ground heat flux and outgoing longwave radiation, i.e. $-(K_i + \varsigma_{\mathrm{L},i}(\psi_{\mathrm{sky},i}L_{\mathrm{sky},i}^{\downarrow} +$

$L_{\mathrm{env},i}^{\downarrow}) + H_i)/\lambda_1)$. Matrices C, D and E stem from Eq. (39) (see Suter (2019) for more details). After rearranging and substitution of (40) into (41) we obtain

$$(C + DA^{-1}B)\frac{\mathrm{d}\mathbf{T}}{\mathrm{d}t} = EA^{-1}B\mathbf{T} + EA^{-1}\mathbf{b},\tag{42}$$



where $^{-1}$ indicates matrix inversion. Note that it has been assumed here that $\mathbf{b}$ does not depend on time for simplicity. Time integration is done with a fully implicit backward Euler step. The matrices involved in this scheme are of the size $(k+1)^2$ and matrix inversion can become expensive for walls with a large number of layers, especially considering that the calculation has to be performed for each facet individually. The pre-calculation of $\mathrm{A}^{-1}$ is possible if all facets have the same number of layers and one remaining matrix division is required for the time integration. The surface energy balance has been validated in Suter (2019).

## 3.6 Surface energy and water budget of vegetated surfaces

The surface energy budget of vegetated surfaces follows the description in section 3.5 very closely. Since no tall vegetation is currently being considered, we represented vegetated surfaces as normal facets. The facet properties can be chosen accordingly, e.g. green roofs are often thicker than conventional roofs and vegetation albedo differs from building materials. To represent the additional roughness introduced by vegetation $z_0$ and $z_{0h}$ can be adapted.

Additional to the terms in Eq (32) the latent heat flux $E$ is being introduced:

$$K_i + L_i(T_0) + H_i(T_0) + E_i(T_0) + G_i(T_0) = 0. \tag{43}$$

The values of $E$ are the time mean fluxes provided by the wall function for moisture and are based on the surface properties from the previous time-step.

The soil moisture content is then updated according to the water balance of facets:

$$W_{\mathrm{G},i}^{n+1} = \max\left(0, W_{\mathrm{G},i}^n + \frac{E_i}{L_{\mathrm{v}}}\Delta t_{\mathrm{E}}\right), \tag{44}$$

where $L_{\mathrm{v}}$ is the latent heat of vaporization. Furthermore the terms for plant canopy resistances, soil resistance and relative humidity at soil level used in the wall function for moisture (2.4.2) are adjusted based on the new values of the green roof temperature, soil moisture and radiation.

## 3.7 Integration into LES

A schematic of the implementation and the order of several model routines are shown in Figure 6. Every time-step (except the initial one) starts with applying the conditions at the lateral boundaries, e.g. periodicity. Then the effects of wet thermodynamics, such as temperature changes due to evaporating and condensing water, are considered. In the advection step quantities are transported according to the resolved velocity field. The transport due to the non-resolved sub-grid processes is modelled directly after. Followed by additional forcings such as the Coriolis effect. Finally, the top and bottom boundary conditions are applied and the immersed boundary method ensures that there is no wind and transport into buildings. At this point the momentum, temperature and moisture fluxes from the wall functions are calculated for all surfaces based on their surfaces properties. The fluxes are then added to the prognostic equations (2) & (3) before time integration is done and a new time-step starts. The calculation of the energy and water budget is not done at every LES time-step ($t^n$), since the time scales associated with the





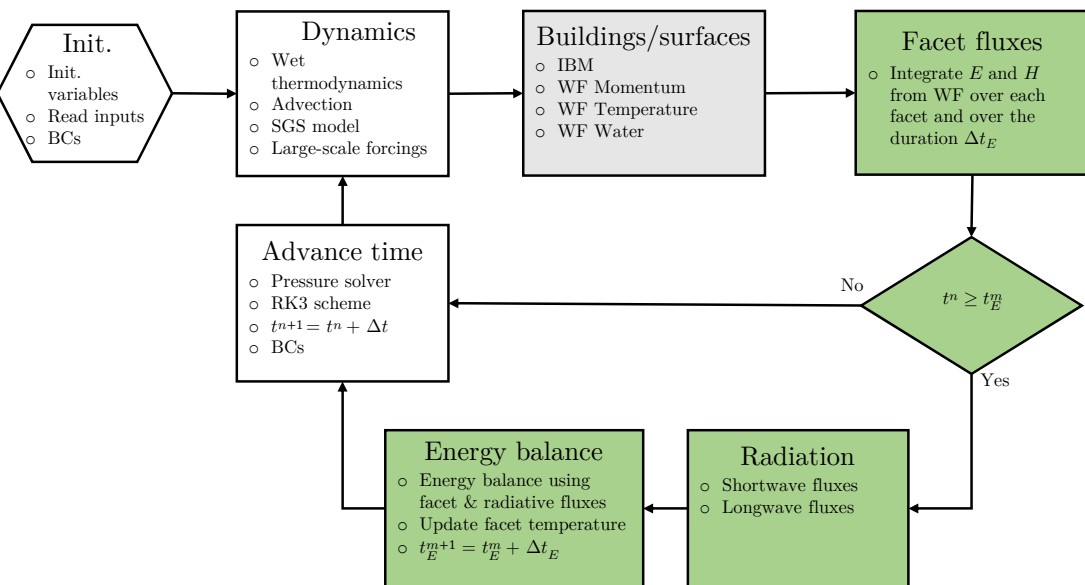

**Figure 6.** Schematic of the main routines in uDALES, unshaded boxes refer to DALES core routines (see Heus et al. (2010)). The grey box indicates the routines that are used when considering buildings. The green boxes refer to the energy balance routines

walls are generally much larger than the atmospheric ones. The routines to update the energy balance are thus only invoked after several $\Delta t$. In every iteration of the main routines the wall fluxes are thus accumulated. Furthermore the walls may extend across processor boundaries. All processors know about the facet properties and calculate the local wall fluxes according to the state of the local fluid cells. For the facet energy balance the processor average has to be determined. If between two energy balance time-steps ($\Delta t_{\mathrm{E}}$) $p$ LES time-steps ($\Delta t$) occur, and the facet area ($A$) is split across $l$ processors, the mean latent and sensible heat fluxes during that time period can be calculated as

$$E_i = \frac{1}{A\Delta t_{\mathrm{E}}} \sum_{o=1}^{p} \sum_{k=1}^{l} A_k E_i^o \Delta t^o, \tag{45}$$

$$H_i = \frac{1}{A\Delta t_{\mathrm{E}}} \sum_{o=1}^{p} \sum_{k=1}^{l} A_k H_i^o \Delta t^o. \tag{46}$$

At every $t_E$ the longwave and shortwave budgets are updated, the water budget is calculated according to Eq. (44) using the values obtained from Eq. (45) & (46). The new facet temperatures are calculated following Eq. (42) and subsequently the facet properties are updated accordingly. These facet average fluxes and the calculation of the energy balance is done on a single processor; for this purpose the information is gathered via MPI and the updated facet properties are redistributed afterwards.



## 4 Test cases

### 4.1 Validation using DAPPLE windtunnel experiments

The dynamic core of uDALES (DALES) has been validated extensively and used as part of several atmospheric intercomparison studies (Heus et al., 2010). The adapted version of the code with the IBM implemented has been validated against both wind and water tunnel data for idealised geometries (Tomas et al., 2016; Tomas, 2016). A further validation of uDALES is presented for its application to realistic urban morphologies and passive scalar dispersion under neutral dynamic conditions using wind-tunnel data from the Dispersion of Air Pollution and its Penetration into the Local Environment (DAPPLE) project.

DAPPLE was a large, multi-disciplinary study of both urban meteorology and pollution dispersion that consisted of field measurements, wind tunnel modelling and computational simulations (Arnold et al., 2004). The case study area was the region surrounding the junction of Marylebone Road and Gloucester Place in Central London, United Kingdom. Wind tunnel experiments were conducted over the case study region as part of this project (alongside passive scalar releases in the field; Carpentieri et al., 2009, 2012). The wind tunnel model was set-up at a 200:1 scale, the freestream velocity was 2.5 m s$^{-1}$, the wind direction was south-westerly and a turbulent neutral boundary layer was developed upstream of the urban geometry using Irwin spires and 2-D roughness elements. Xie and Castro (2009) conducted an LES study that reproduced these wind-tunnel results with good agreement.

### 4.1.1 Simulation set-up

The main simulation uses inflow-outflow boundary conditions in the streamwise direction and is driven through a precursor (driver) simulation. Details of the two simulations are provided in Table 1. A diagrammatic view of the modelled urban morphology is shown in figure 7, reproducing the model used in the wind tunnel experiment at full scale. The markers and the axis $x_r$ in figure 7 indicate the location of scalar point sources and measuring positions. The case study morphology is rotated such that the south-westerly direction is aligned with the $x_D$-axis. The mean building height, $h_m$, is 22 m and the reference velocity, $u_{ref}$, is 2.5 m s$^{-1}$.

The use of a precursor simulation allows for the incoming boundary layer to be defined independently of the main simulation (which can be particularly useful when reproducing wind tunnel experiments or studying transitional flows). The driver simulation is set up to reproduce the boundary layer dynamics obtained upwind of the wind tunnel morphology in the experiment of Cheng and Robins (2004). A staggered cubic array was used in the driver simulation, with a mean block height of 4m. The height and packing density (0.25) was obtained from the mean velocity profile using the relationship defined by Macdonald et al. (1998). Figure 8 compares the mean and turbulent statistics produced in the driver simulation against the incoming profiles from the wind tunnel. A good agreement is shown indicating that the inlet to the main simulation closely matches that of the wind tunnel data. The driver simulation had to be run-up to reach a statistical steady state prior to obtaining converged statistics (run-up times and averaging periods given in Table 1).



**Table 1.** Simulation set-up details for driver and main simulations used in the uDALES validation. Boundary conditions (BC) defined for the $x$- and $y$-directions.

|  | Simulation | |
| --- | --- | --- |
|  | Driver simulation | Main simulation |
| Grid size | $200{\times}400{\times}100$ | $450{\times}400{\times}100$ |
| Domain size [m] | $400{\times}800{\times}200$ | $900{\times}800{\times}200$ |
| $x$ momentum BCs | Periodic | Inflow-outflow |
| $y$ momentum BCs | Periodic | Periodic |
| $x$ scalars BCs | - | Inflow-Outflow |
| $y$ scalar BCs | - | Inflow-Outflow |
| Flow forcing | $F_x = 0.000125$ m s$^{-2}$ | Inlet from driver sim. |
| Run-up time [s] | 50,400 | - |
| Averaging period [s] | - | 25,200 |

### 4.1.2 Results

Mean and turbulent flow and scalar statistics obtained in uDALES are compared to those of the wind tunnel experiments.
Figure 9 compares the mean and turbulent vertical profiles obtained at position R (note all variables are rotated for alignment
570 with the axes $x$ and $y$; refer to Figure 7). Position R is situated within a large intersection and directly downwind of an elevated
spire. The simulation illustrates a good agreement between all three mean velocity profiles. The higher order statistics are also
shown to generally provide a good agreement, with the most significant discrepancy shown to be the relatively larger peaks in
the Reynolds stress profiles at $z = 2h_m$. These peaks are related to the wake of the upwind spire. Capturing the form of this
spire is challenging due to its oblique angle upon the Cartesian grid (refer to Figure 7) and as such both its frontal area and
575 vertical form may lead to the relatively larger fluxes over this region (the dimensions of the spire also had to be approximated
due to a lack of available data). Similar agreement was obtained over 8 other positions within the modelled domains (the red
crosses in Figure 7).

Validating the model against a point source release of a passive scalar acts to validate the integrative effects of the modelled
flow field (both within and above the UCL) and the treatment of scalars in the model (in particular the $\kappa$-advection scheme).
580 Figure 10 displays the normalised pollutant concentration and variance along the axis $x_r$ (as defined in Figure 7). The
simulation is shown to provide a very close agreement to the results obtained in the wind tunnel experiment both in terms of
the mean concentration and the normalised root mean square concentration. This finding validates the application of uDALES
to model pollution dispersion within realistic urban morphologies.



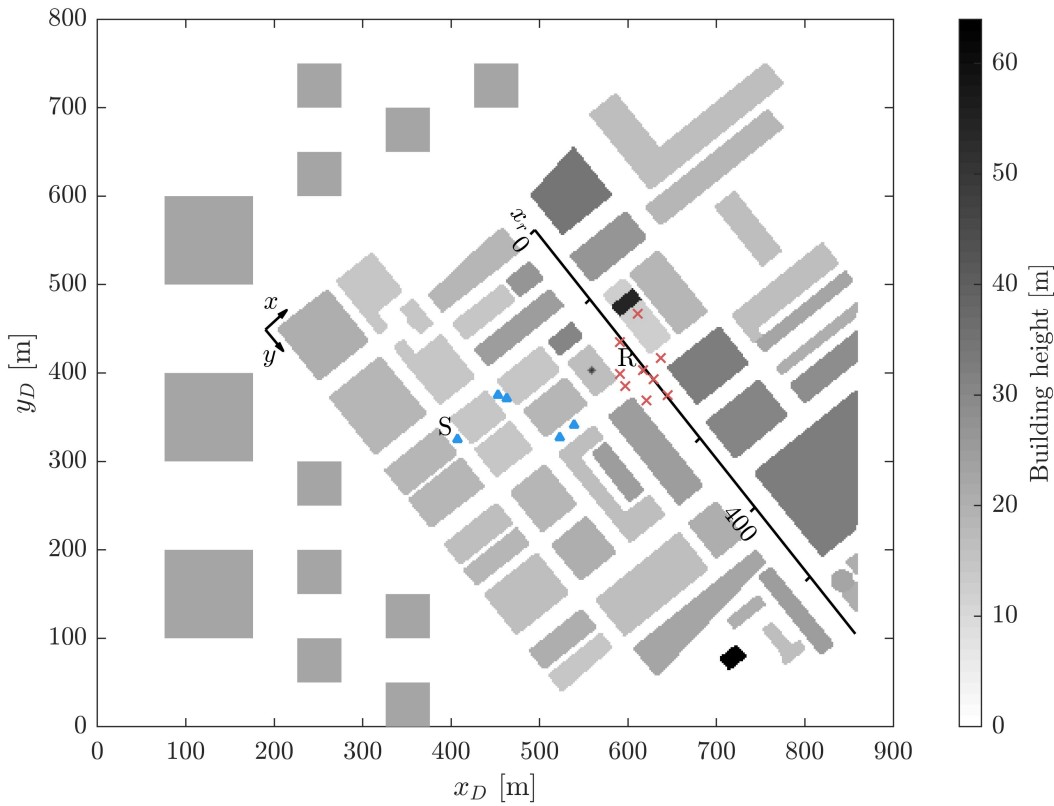

**Figure 7.** Diagrammatic view of the urban morphology used in the main simulation. The positions of the measuring points around the main intersection are denoted by red crosses (point R used in this validation) and the position of the scalar releases are denoted by blue triangles (release point S discussed here). An additional horizontal axis, $x_r$, is used to plot a profile of the spanwise pollutant concentration along the Marylebone Road. The axes $x$ and $y$ are aligned with the Marylebone Road and used in the results section.

## 4.2 Verification of surface energy balance

585 To study the behaviour of the surface energy balance in uDALES a simple case has been set up. The boundary conditions are periodic in $x$ and $y$ and a zero-flux boundary is imposed at the top. The geometry is shown in Figure 11a and the scenario parameters are listed in table 2.

The focus here lies in the interaction of LES, wall function and surface energy balance. The sensible heat flux $H$ is calculated by the wall function at every LES time-step ($O(1s)$) and the heat is added/removed from the fluid. This flux is integrated

590 between energy balance time-steps by using Eq. (46) with $\Delta t_{\mathrm{E}} = 300$ s. For the boundary conditions chosen, the temperature change of the fluid has to equal the total energy flux from the surface.




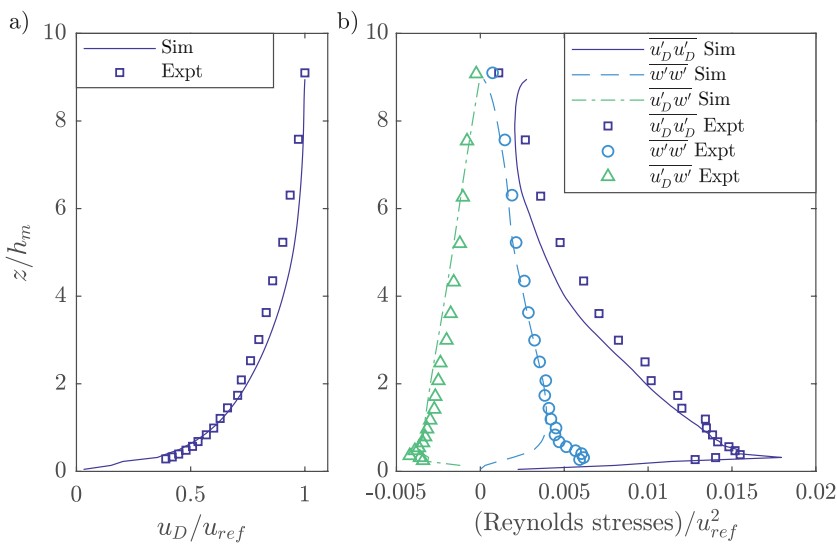

**Figure 8.** (a) Mean and (b) turbulent profiles of the precursor simulation and the boundary layer developed before the urban area in the wind tunnel experiments of the DAPPLE project. Note the use of subscript $D$ to denote alignment with the axes $x_D$ and $y_D$ (see Figure 7)

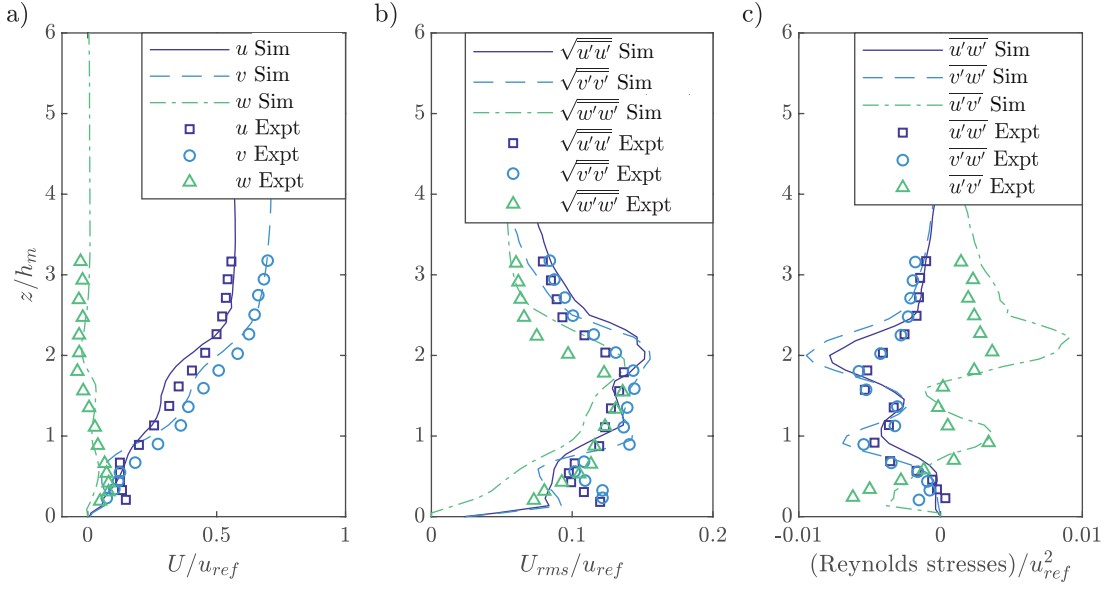

**Figure 9.** Vertical profiles of the mean velocities (a), root-mean-square velocities (b) and Reynolds stresses (c) at position R. Plotted against the wind tunnel experiment results of the DAPPLE project.



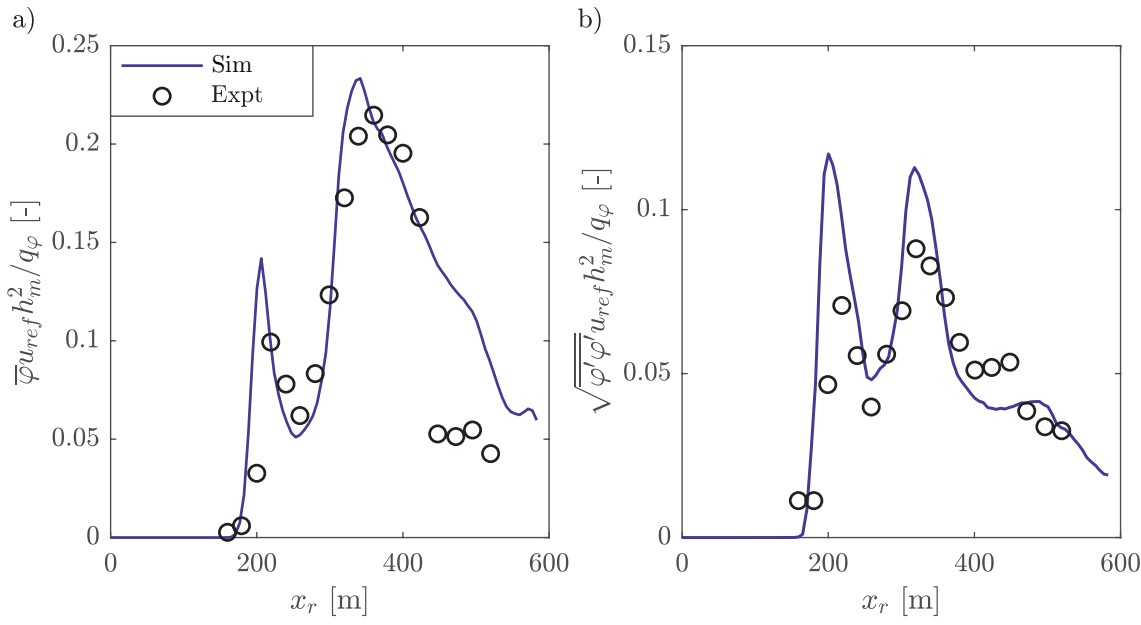

**Figure 10.** Comparison of the (a) normalised scalar concentrations $\varphi$ and (b) root-mean-square concentrations of the wind tunnel experiments of DAPPLE and simulation along the axis $x_r$ for a point source scalar release at position S.

**Table 2.** Scenario parameters.

| Property | Value | Property | Value |
|---|---|---|---|
| Domain size | $72 \times 72 \times 72$ m | Facet emissivity | 0.85 |
| Solar elevation | $45°$ | Nr. of wall layers | 3 |
| Solar azimuth | $90°$, E | Wall thickness | $3 \times 0.1$ m |
| $I$ | $750$ W m$^{-2}$ | Road thickness | $3 \times 0.33$ m |
| $D_{\mathrm{sky}}$ | $100$ W m$^{-2}$ | Freestream velocity U | $2$ m s$^{-1}$ |
| $L^{\downarrow}_{\mathrm{sky}}$ | $200$ W m$^{-2}$ | Building temp. | $301$ K |
| Facet albedo | 0.5 | Initial air temp. | $300$ K |

Figure 11c shows the temperature evolution of all facets. It is evident that the temperatures will largely be influenced by the solar radiation, accordingly east-facing facets are the warmest. Furthermore, one of the roof facets was given green roof properties, resulting in lower surface temperatures and it clearly stands apart from the other roofs.

Figure 11d shows the surface fluxes of a single road facet. Net shortwave radiation $K$ is unchanged throughout the simulation and $L$ only varies slightly since most facets don't experience a large surface temperature change. The sensible heat flux $H$



(a)

(b)

(c)

(d)

**Figure 11.** (a) Geometry of the surface energy balance test case. (b) Comparison between uDALES and MTEB of the total energy change of the air over time. (c) Temperature evolution of all building facets. (d) Energy fluxes of a single floor facet.





shows fluctuations due to the turbulent nature of the flow. Figure 11d also demonstrates that the net flux $(K + L + H)$ matches
the ground heat flux $(\lambda \frac{\partial T_0}{\partial z})$ predicted by the energy balance, verifying that the energy balance is correctly linked to the wall
functions.

The case presented in this section is very idealised and is thus well suited to also be studied with the urban energy balance
model MTEB (Suter et al., 2017). Most of the parameters can be carried over directly. Solar/radiation properties, air temperature
and surface properties are identical to table 2. To obtain the idealised canyon geometry in MTEB we can consider the total
area covered by buildings and roads. This results in a building width $b = 72 \cdot (14 \cdot 8^2)/(72^2) = 12.44$ m, canyon width $l =
72 - 12.44 = 59.56$ m and a building height $h = 8$ m. The initial surface temperatures in MTEB were chosen to be the mean of
the corresponding facets in uDALES. The energy balance of the entire domain is determined by the balance of the net shortwave
and net longwave at the top. The effective albedo of the topography can be calculated in uDALES since the absorbed shortwave
of all facets and the incoming shortwave at the top is known. Whereas all the facets have an albedo of 0.5 the resulting effective
albedo is $\approx 0.35$ due to radiation trapping. In MTEB the resulting effective albedo is $0.37$ and thus in good agreement. The net
longwave on the other hand evolves during the simulations since the surface temperatures change. Figure 11b shows the total
energy in the system over time for both the uDALES and MTEB runs. The canyon in MTEB does not have a heat capacity,
this leads to an initial jump of energy in the system at the start of the simulation. Over time the two curves agree well. Since
the two models are completely independent this is a good indication that the surface scheme in uDALES produces plausible
results.

### 4.3 East Side demo

To highlight the capabilities of the model developed in this dissertation, this chapter discusses the effect the installation of a
green roof on a single building has on the local microclimate. The test area is the Imperial College campus in South Kensington,
London. The green roof was installed on the "Eastside" building. The meteorological conditions were those of 21st June 2017,
the hottest day in Heathrow, London in 2017. Air temperatures reached 35°C at around 16:00h, wind was easterly and the sun
was at an elevation of 36° and an azimuth of 130° from north, counterclockwise. Before the simulation the morphology input
files had to be prepared. Small gaps along the building walls were closed, small objects and any pixel with only one neighbour
removed. Finally, depressions inside buildings (courtyards, etc.) were filled. Small depressions should be removed since the
circulation in such enclosures cannot be accurately represented by the LES model. The buildings were thereafter sliced, the
ground was populated with road facets and a bounding wall was created. The results of this division can be seen in Figure
12a. View factors between the facets were then calculated and stored. For simplicity all facets were assumed to have identical
properties shown in Table 3.

The horizontal domain size is 960 m×480 m with a resolution of 2.5 m in each dimension. Vertically the grid was 2.5
m within the canopy and gradually stretched above to a total height of 768 m, resulting in $384 \times 192 \times 192$ grid points. The
simulation (10 h) was run on 16 CPUs on a single workstation for $\approx 125$ hours. The lateral boundary conditions were periodic
and 7 hours of neutral conditions were simulated to allow the development of turbulence in the domain. The facets were
initialised with outside temperatures corresponding to their radiative equilibrium and a linear internal temperature profile. The





| Wall properties | | | | | | | |
|---|---|---|---|---|---|---|---|
| $z_0$ [m] | $z_{0h}$ [m] | $\alpha$ | $\epsilon$ | d [m] | $\rho c_{\mathrm{p}}$ [$\frac{\mathrm{J}}{\mathrm{m}^3\mathrm{K}}$] | $\lambda$ [$\frac{\mathrm{W}}{\mathrm{mK}}$] | $\varkappa$ [$\frac{\mathrm{m}^2}{\mathrm{s}}$] |
| 0.05 | 0.00035 | 0.5 | 0.85 | 3×0.12 | $2.5 \cdot 10^6$ | 1 | $4 \cdot 10^{-7}$ |

**Table 3.** Wall properties of facets in the "Eastside" simulation. The defined material properties corresponds to concrete (Asadi et al., 2018; The Engineering Toolbox, 2018).

radiative equilibrium temperatures are generally too hot and for applied studies of the urban environment it is important to consider these initial conditions carefully. It was assumed that unlimited water for evaporation was available in the soil. The

atmosphere was set to a uniform temperature of 300K. The spin-up was then continued for an additional hour to allow the development of thermal effects and convection. After 8 hours two simulations with and without green roof on "Eastside" were run for an additional hour and then compared.

It is to be noted that due to the nature of the simulations differences in velocities and scalar quantities can be caused by a number of effects and not just the direct presence of the green roof. The slight perturbation of the initial and boundary conditions

can lead to substantial differences after some time and due to the strongly coupled nature of the system complex flow patterns can emerge and turbulent structures can persist for a prolonged period. Furthermore transient processes are present, driven by the slow warming of the entire domain. The mean quantities of the simulations will consequently evolve over time and ideally ensembles runs would be studied for statistics.

While cooler roof temperatures can be significant for indoor climate, most of the outdoor activities take place on street level.

LES is ideal to study how the air from above the roofs is transported and mixed throughout the atmosphere.

Figure 12b shows the difference between one-hour means of the specific humidity field from a simulation with and without the green roof. We can observe clear downward mixing in the building wake. Since there is no other humidity source in the domain, the humidity difference can be attributed entirely to the presence of the green roof. On street level a mean increase in specific humidity of $0.01\,\mathrm{g\,kg^{-1}} - 0.1\,\mathrm{g\,kg^{-1}}$ can be observed. This corresponds to a almost insignificant difference in relative

humidity, since at $30°$C a relative humidity change of 1% equals a specific humidity change of $\approx 0.3\,\mathrm{g\,kg^{-1}}$.

Facet temperatures at the end of the simulation are visualised in Figure 13a. Temperature maxima occur in west and south facing walls exposed to the sun, with temperatures up to 335 K or $62°$C. North and East facing facets on the other hand are close to the air temperature of $\approx 303$ K (not visible). While roofs are exposed to a large shortwave forcing, the sensible heat flux is usually larger than for vertical surfaces due to the higher tangential wind velocities, leading to somewhat cooler surfaces. The

green roof stands clearly apart from other roofs; the temperature remains also very close to the air temperature even though it is directly exposed to the sun. The resulting latent heat flux from soil and vegetation leads to a surface temperature reduction of $\approx 10$ K. This can also be seen when comparing the internal temperature profiles of a simulation with and without green roof in Figure 13b. The temperature profiles between the four resolved points were reconstructed following Eq. 36. From Figure 13b it is also evident that the initial temperatures of the roofs were chosen too high and the roof has been cooling down over time

highlighting the importance of accurate initial conditions. Yet, this is nearly impossible to avoid since the internal temperature





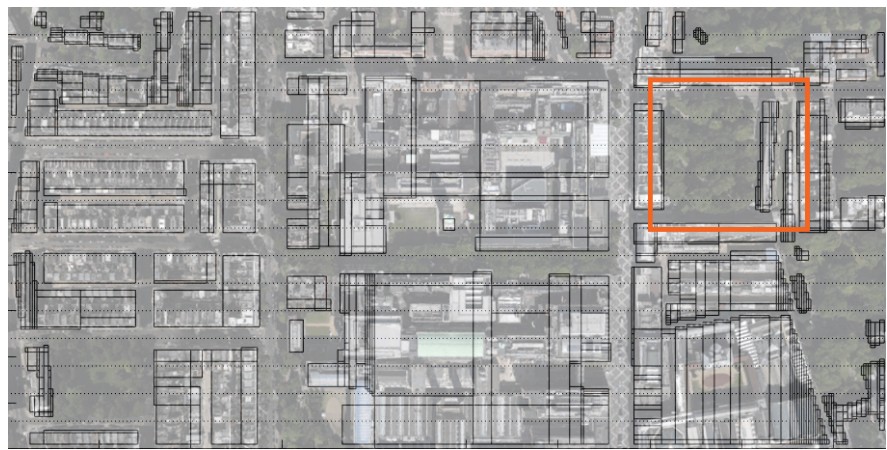

(a) Imagery ©2021 Google, Imagery ©2021 bluesky, CNES / Airbus, Getmapping plc, Infoterra Ltd & Bluesky, Maxar Technologies, The GeoInformation Group, Map data ©2021 Google

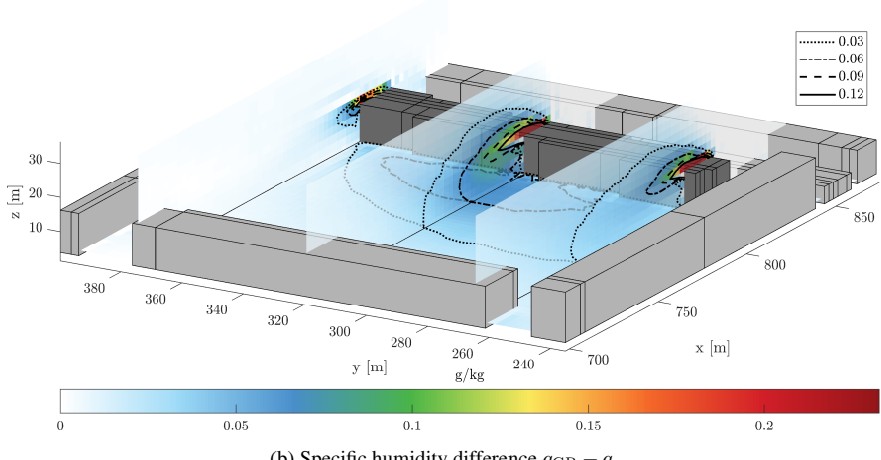

(b) Specific humidity difference $q_{GR} - q$

**Figure 12.** (a) Morphology of South Kensington Campus and surroundings divided into building blocks and overlaid over aerial image. (b) Three vertical and one horizontal slices through difference between one-hour means of simulations with ($q_{GR}$) and without green roof ($q$). "Eastside" building coloured in darker grey.



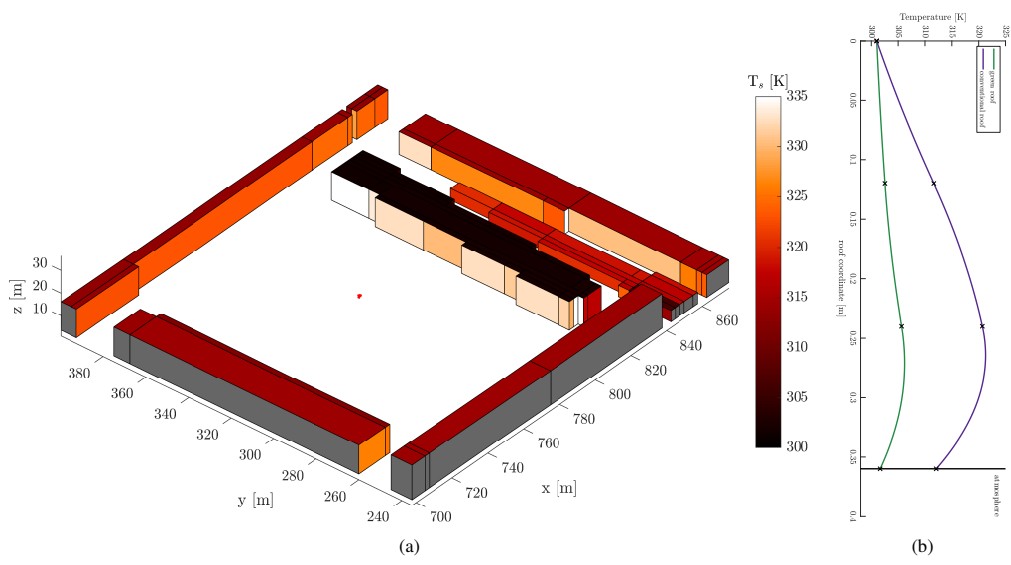

(a)                                                                (b)

**Figure 13.** (a) "Eastside" building and surrounding with building facets coloured based on their surface temperature. Floors are not colored for clarity, grey surfaces are building internal. View from direction of the sun. (b) Internal roof temperature profiles for simulations with and without green roof on the "Eastside" building. The roof coordinate is the reverse of the depth $z$ defined in Fig 5. It points from the roof interior to the outside.

profiles are usually not known and depend on past environmental conditions due to their long associated timescale.

Figure 14 shows the temperature evolution and surface energy fluxes of the "Eastside" roof for both cases. The surface temperatures of both the conventional and green roof decrease over the course of the simulation. The temperature reduction is

largely driven by the latent heat flux $(E)$ and by the sensible heat flux $(H)$ for the green and the conventional roof, respectively. For the green roof the radiative fluxes on the facet surface almost balance. Net shortwave radiation $(K_{\mathrm{net}})$ does not change during the simulation, since the solar position does not change. There is a minute shift in incoming longwave radiation $(L_{\mathrm{in}})$ due to the temperature change of surrounding facets. However, only a tiny fraction of the field of view of a roof facet on "Eastside" is occupied by other facets; the sky constitutes the largest part. The magnitude of the emitted longwave $(L_{\mathrm{out}})$ follows the

Stefan-Boltzmann law and decreases along with the facet temperature. The remaining three fluxes are the surface conductive heat flux $(\lambda \frac{\partial T_{\mathrm{s}}}{\partial z})$, the sensible and the latent heat flux. These three fluxes vary more substantially on short timescales. The surface energy balance was calculated every ten seconds and thus the turbulent fluctuations of the sensible and latent heat flux can still be distinguished. For the green roof the sensible heat flux is relatively small, since the temperature difference to the air is minor. The net energy loss that leads to the cooling is thus caused by evapotranspiration from the vegetation. The latent heat

flux is also a function of the temperature difference to the air and we can see a decline over time. For the conventional roof the radiative fluxes are close to balance as well, but no evapotranspiration is possible and $E = 0$. The strong temperature difference initially leads to very large sensible heat fluxes. The large temperature gradient within the roof further causes substantial heat



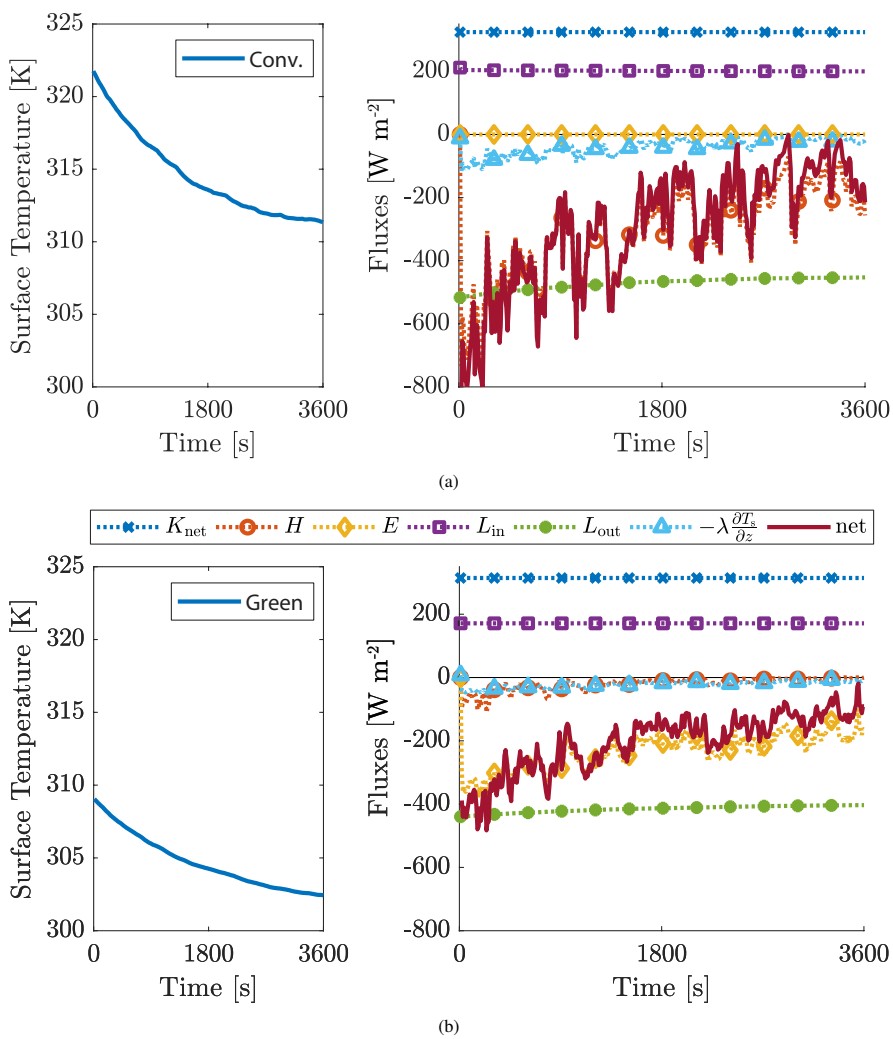

**Figure 14.** (a) temperature evolution of a green roof facet and the corresponding surface energy fluxes. (b) Temperature evolution of a conventional roof facet and the corresponding surface energy fluxes. The markers are for the sake of clarity, the data was plotted every 10 s and the markers do not correspond to particular data points.

conduction into the roof. As the surface cools down these fluxes also taper off.

## 680  5   Concluding remarks

The new urban LES model uDALES was presented in this article. The inclusion of explicit representation of buildings as well as energy, vegetation and chemistry processes into a high-resolution atmospheric LES opens a multitude of new possibilities



in terms of simulating urban air quality and the urban climate. The model parts have been verified or validated and results are shown here or referenced accordingly: in section 2.4.1 the implementation of the wall function is verified against simulations

by Cai (2012a), where we find good agreement in temperature values and distribution for flow over a canyon-like geometry; the scalar transport, building representation, turbulence and advection models are validated against wind tunnel experiments in section 4.1. Modelled wind velocities and Reynolds stresses soundly agree with the experiments and the scalar concentrations match very well; the general performance of the implemented surface energy balance is verified in section 4.2 by comparing with an urban energy balance model.

The validation and verification cases already hint at a number of possible applications of uDALES and in section 4.3 a showcase of the full model capapilities is demonstrated. uDALES has been used to Improved simulations of the urban micro-climate can be performed with uDALES, in particular to study: the effect of urban vegetation on the outdoor climate and the energy budget of buildings; the attribution of the urban heat island effect to various processes such as radiation trapping or heat storage in the built environment; pollutant transport and reaction with very high resolutions to investigate hotspots and

real-time pedestrian exposure

The effect of urban trees is currently widely studied and initial steps have been undertaken to include tall vegetation into uDALES (Grylls and van Reeuwijk, 2021). Trees can have a significant influence on pedestrian-level air quality and temperature; their interaction with a variety of processes such as radiation, latent heat and turbulence generation demand a very careful consideration, however. The huge parameter space of cities make general conclusions from simulations difficult. It is therefore

of great interest to be able to simulate a variety of similar cases, where as few parameters change as possible. The modelling of buildings on a Cartesian grid with immersed boundaries allows an effortless set-up of idealised urban layouts. uDALES is currently being further developed to incorporate automatically generated urban landscapes, with which one can study the effects of various building configurations linked to parameters such as plan and frontal area density (Sützl et al., 2021). A systematic study of building patterns addresses the complexity gap in current urban studies between idealised building geometries and

case specific real urban geometry.

*Code availability.* The code of uDALES has been published in the Journal of Open Source Software (Grylls et al., 2021).

*Data availability.* All the output data and scripts to produce the relevant figures 2 and 7 - 14 are temporarily hosted on: https://polybox.ethz.ch/index.php/s/AUraSzQgOeMVWdD and will in due time be moved to a permanent Zenodo repository. The model input for the comparison with Cai (2012a, section 2.4.1), the test of the energy balance (section 4.2) and the show-case (section 4.3) have also been included. A general

guide on how to set-up the model and examples are available on the code repository.



*Author contributions.*  IS authored the article and prepared tables and figures. IS initially set-up and validated DALES on Imperial College infrastructure, adopted the model development by Tomas et al. (2015) and added and validated facets, wall-functions, wall energy and water balances. TG extended the model with emission and nitrogen oxide/ozone chemistry capabilities, as well as inflow boundary conditions. TG and BS conducted the comparison with DAPPLE and authored the corresponding section. MvR acquired funding, supervised and assisted in
the projects. All authors edited the paper text.

*Competing interests.*  "The authors declare that they have no conflict of interest."

*Acknowledgements.*  The authors wish to acknowledge the support of the European Institute of Technology, Climate KIC Innovation project Blue Green Dream; EPSRC Centre for Doctoral Training in Sustainable Civil Engineering (grant reference EP/L016826/1); EPSRC Mathematics of Planet Earth Centre for Doctoral Training (Grant Reference EP/L016613/1); and the computational resources for this work were
provided by the Imperial College high-performance-computing facilities.

    We would like to extend our gratitude to Prof. Harm Jonker and Dr. Jasper Tomas who thoroughly introduced us to the DALES model, and to David Meyer and Sam Owens for their contribution to the code availability and maintenance.





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
