# Peer review of "uDALES 1.0.0: a large-eddy-simulation model for urban environments"

_Geoscientific Model Development, 2021_

## Author Response (AR1)

**Reply to CC1:**

I would like to congratulate the authors on their excellent work. Just a minor comment on the introcucion of the paper: The references to Maronga et al. (2015) and Resler et al. (2017) could eventually be replaced by the more recent references Maronga et al, 2020 (https://doi.org/10.5194/gmd-13-1335-2020) and Resler et al, 2021 (https://doi.org/10.5194/gmd-14-4797-2021) .

Dear Dr. Forkel
Thank you for the recommended updates on the references. We changed the references to PALM
(*Overview of the PALM model system 6.0*, https://doi.org/10.5194/gmd-13-1335-2020 and *Validation of the PALM model system 6.0 in a real urban environment: a case study in Dejvice, Prague, the Czech Republic*, https://doi.org/10.5194/gmd-14-4797-2021)

**Reply to RC1:**

**General comments:**

This manuscript provides a technical documentation for the open-source large-eddy simulation (LES) code uDALES which has been augmented with capabilities to handle arbitrary urban morphologies and to model the local evolution of surface energy budgets and their interaction with the flow. The documentation also includes the implementation of a simple chemistry model. Overall, I'm very pleased with the model development part, which has been documented, for the most part, with sufficient care and clarity. This paper will be an important reference for the uDALES community. My primary concerns relate to the validation practices which do not meet modern scientific standards. With the condition that these validation short-comings are corrected, along with few other minor comments below, I gladly recommend this manuscript for publication.

We thank reviewer #1 for their constructive comments which have made the paper more accessible and rigorous, especially in regards to the validation practices.

The main changes made in this revision are:

1. Further explanation on the technical elements of the paper;
2. Detailed validation metrics for the DAPPLE case;
3. Substantial improvement of the East-side demo:
   a. Reran the experiment with the use of a driver simulation rather than a periodic simulation in order to prevent heating up of the air;
   b. This allows the simulation time to be 8 hours rather than one hour.
4. Fluxes are now presented conforming to atmospheric convention.
5. A thorough check of all simulation cases in the repository, with updated input and run-options.

We respond to the specific points made by the reviewer below.

**Specific comments:**

Comment 1: P1, l17: "… since these structures channel high-momentum air downward to ground level". This keeps popping up in the context of urban boundary layer (UBL) flows, yet I still have not seen proper evidence for it, let alone an explanation what is the physical mechanims for high-rise buildings creating *high-momentum* flow structures towards the ground. The downwash in front of the building or the wake turbulence behind it should not warrant such a simplistic characterization of wind conditions near the ground. I checked (Blocken et al. 2015) and it appears that the introduction, at least, provides only anecdotal evidence. If the evidence lies in some of the references within Blocken 2015, please point to it directly or refrase the original statement so that the readers will not continue to spread this claim as a fact.

Authors reply: We thank the reviewer for pointing this out. By no means we want to trivialise the flow structures in complex urban environments. The flow patterns at ground level of high-rise buildings will always be determined by the surrounding geometry,

topography and meteorological conditions, indeed this is the reason why we bring up high-rise buildings as example for the use of CFD studies.

The study of wind conditions at pedestrian level and observation that high-rise buildings generally increase mean wind speed and gustiness around the corners of tall buildings goes back to the 1960s, with a number of important publications following the International Conference on Wind Effects on Buildings and Structures in Heathrow, London (e.g. Lawson and Penwarden, 1975, Isymov and Davenport, 1975).

As the wind speed in the boundary layer increases with height, tall obstacles tend to intercept the stronger winds high up and redirect them down to ground level, causing accelerated flow zones in front and around the corners of the building (ASCE, 2004). The amplification of ground level wind speed is often measured in terms of an amplification factor, that compares the existing geometry with/without the tall building (Blocken et al., 2003). While improved building design can alleviate some of these effects (for example by using a podium shaped base or a floor plan area that decreases with height), the building height relative to the height of the surrounding buildings is always a decisive factor (ASCE, 2011).

Publications by the American Society of Civil Engineers (ASCE 2004, 2011) were added as references since these provide an excellent overview of the most important works, research progress and detailed explanation of the physical mechanisms influencing pedestrian level wind comfort. We also added "can" to the sentence to account for the fact that ground level flow always depends on the individual geometries.

Comment 2: P5, l122: The pressure field is solved via FFT (which, by default, requires horizontally periodic boundary conditions), but some of the lateral boundary conditions described in Section 2.4.2 and Figure 1 are not horizontally periodic. Please specify how the pressure field is solved in those bc configurations.

Authors reply: When using inflow/outflow boundary conditions, the y direction is periodic so is still solved using FFT, but x and z are solved using cyclic reduction. The inflow boundary condition for pressure is Neumann and the outflow is Dirichlet. The bottom and top are both Neumann boundary conditions. We have clarified this in section '2.2 Immersed boundary method'.

Comment 3: P7, l173: Please clarify are the dimensions (Lx x Ly x Lz)? Is the order of grid resolutions now correct? It's best to document with clarity so you don't leave the reader guessing.

Authors reply: To avoid any confusion we now clearly state the size of the model domain in number of cells & size of cells and total extent in metres

Comment 4: P7, l183: You claim to present quantitative agreement, but do not provide any quantity. Eye-balling the temperature contours from Figure 2 is not a satisfying validation strategy. Please provide a transparent validation metric for the comparison which other

researchers in the field can compare their results to or reduce the comparison to a qualitative one.

Authors reply: We thank the reviewer for noticing this – we intended to state the agreement was qualitative. We reproduce figures of Cai without having the original data and cannot carry out a quantitative comparison. The text in '2.4.1 Wall-functions for momentum and temperature' was changed to make clear that the comparison is qualitative.

Comment 5: P12, l295: Section 3.1 needs to be improved. The description of how surfaces within the discretized model are handled as facets is not clear. It is very difficult to understand how the LES grid relates to the facets used by the UEB model. Given the IBM implementation, the surfaces of all building blocks are made up of potentially a large number of cell faces, so are they grouped to form facets? It appears so but it's hard to get a clear picture of the method. What happens when the buildings are diagonal with respect to the LES grid? The Figure 4 provides some help but not much. Figure 11a is much better so you should make better use of it. Now it comes at the very end of the section, which is a struggle to reach with the current description.

Authors reply: The process is to generate 'blocks' first. These blocks represent e.g. entire buildings and have to conform to the LES grid. In case a block does not touch any other blocks, the facets would correspond to the 5 faces of said block (the face touching the ground is ignored). However, if two blocks touch, one has to subdivide one or both blocks to ensure that faces are entirely exposed to the fluid (external) or entirely within the touching blocks (internal).

Diagonal buildings cannot be represented directly as the blocks have to conform to the grid. These are represented by stair-case-like patterns, which indeed requires many blocks and therefore many facets. We will be working to remove this limitation in one of the next versions of uDALES.

We extended the introductory part of '3.1 Urban facets' to clarify the process and terminology.

Comment 6: P22, l537: Section 4.1 on Validation. Again, the standards for performing validation against measurements are not sufficient. Comparing vertical velocity and Reynolds stress profiles at one location visually does not constitute validation. And the scalar concentration along a line juxtaposed with measurements alone does not either. Validation metrics are derived for this purpose. What if the authors choose to further improve the numerics of uDALES and perform the same DAPPLE test case again? Will they judge the level of improvement by eye-balling the curves? Of course not. It is true that Xie and Castro (2009) do not carry out validation either, but their paper appears to focus more on establishing sufficient LES modelling criteria for urban flows.

Authors reply: We appreciate the comments on the shortcomings in terms of validation. We now define clear validation metrics: the root-mean-square error (RMSE), fractional bias (FB) and factor of two (FAC2). We have calculated these for the mean profiles of velocity magnitude for the location shown (Figure 9) and 8 other locations, as well as for the mean scalar concentration profile (Figure 10). The validation metrics are shown in Table 2. The

RMSEs range from 0.06 to 0.15 (0.04 for scalar concentration). The FBs are typically slightly below zero, indicating that the simulation had a small tendency to under-predict the velocities. The velocities generally yield high FAC2 scores, with some locations where all simulated velocities lie within a factor two of the experiment data (i.e., FAC2 = 1).

These results are now reported in section '4.1.2 Results' (note the change in notation: measurement point R -> P1; axis for scalar concentration $x_r$ -> $x_s$). The section '4.1 Validation' and '4.1.1. Simulation set-up' were also further expanded to include a more detailed description of the study case and wind-tunnel experiment.

Comment 7: One of their conclusions is that the resolution requirement is ~1 m for this case. Here, validation is attempted and claimed, yet 2 m resolution is used without justification. Thus, please improve the validation section by providing appropriate validation metrics and justifications for the numerical setup.

Authors reply: Indeed Xie and Castro say that ~1m resolution is sufficient for the problem. However, it is not necessarily a requirement. We analysed the subgrid fluxes and they do not exceed 4.5% of the total flux anywhere but in the lowermost cells indicating that the resolution is sufficient to resolve the majority of the energetic turbulent scales of the urban flow field. As always there is a computational cost associated with finer resolutions and we found 2m to be adequate for the problem.

We now mention that the subgrid-scale fluxes are small in '4.1.1 Simulation set-up'

Comment 8: The wordings relating to the validation in Concluding remarks should also be made more scientific so that we do not have to engage in discussions trying to determine the degree of "soundness" in agreements. Verification is a different process and I have no objections with Section 4.2.

Authors reply: We have updated the validation against measurements in section 4.1 and reflect this now in section 5. The text in '5 Concluding remarks' has been changed.

**Technical corrections:**

Comment 9: P17, l432: Only a suggestion: The lower case \sigma_{L,i} does not go well with upper case L in the subscript. If this is not a universally accepted nomenclature, consider using another greek symbol.

Authors reply: We used the fact that we assumed 'emissivity = absorptivity' in the longwave and eliminated the symbol altogether.

In addition, the sign of some of the fluxes in the surface energy balance have been adjusted to better represent the normal conventions in the literature.

Comment 10: P28, l617: "… in this dissertation" This manuscript is likely part of a dissertation but perhaps the wording should be changed here.

Authors reply: Thanks for pointing this out. Changed.

**Reply to RC2:**

This paper presents uDALES, which is a high-resolution, building-resolving large-eddy simulation code for urban microclimate and air quality, to model the local environmental conditions, urban morphology and interaction with the atmospheric boundary layer. In my opinion, the methodology of this paper is scientifically sound.

We thank reviewer #2 for their constructive comments which have made the paper more accessible and rigorous.

The main changes made in this revision are:

1. Further explanation on the technical elements of the paper;
2. Detailed validation metrics for the DAPPLE case;
3. Substantial improvement of the East-side demo:
    a. Reran the experiment with the use of a driver simulation rather than a periodic simulation in order to prevent heating up of the air;
    b. This allows the simulation time to be 8 hours rather than one hour.
4. Fluxes are now presented conforming to atmospheric convention.
5. A thorough check of all simulation cases in the repository, with updated input and run-options.

We respond to the specific points made by the reviewer below.

**Comment 1:** The novelty of this open-source software should be further strengthened. Specifically, compared to OpenFoam and PALM, what is the advantage of this software? Does it also allow modules to be integrated by other users?

**Authors reply:** There are many LES solvers capable of solving flow around obstacles/buildings, such as the open-source code OpenFoam. However, they generally do not include many critical components to determine the urban climate, e.g. humidity or a surface energy balance.

The goals of uDALES and PALM-4U overlap to a large extent. Both models aim to incorporate a large number of processes that determine the urban climate. As such there is no direct advantage over PALM, rather uDALES is an alternative. Compared to global or regional models, there still seems to be a lack of models on an urban scale and we believe having multiple open-source options will benefit everybody. We encourage contributions by other users, and uDALES is set up in a modular fashion that makes it relatively straightforward to add custom modules. We have extended the Introduction to emphasise these points.

**Comment 2:** What extension have you added based on DALES? This should also be clearly stated in this paper.

**Authors reply:** These mainly pertain to the urban surface. We have added a list of differences with DALES at the beginning of the Model description section.

**Comment 3:** The computational time and computational RAM or CPU requirement should also introduce in this paper, as well as the data communication process.

**Authors reply:** A typical use case of 384^3 points at 2 m resolution using a fixed timestep of 1s took about half an hour to simulate 10 minutes using 196 cores. The parallelisation is based on DALES, and we refer to that paper for a more detailed description of the parallelisation.We mention the use of MPI in section '2.2 Method of Solution' and the computational requirements in sections '4.1.1 Simulation set-up' and '4.3 Eastside demo'.

My specific comments are:

**Comment 4:** In equation (2), there is a notation $\theta\_u$; and in equation (3), there is another notation $\theta\_V$. Are they the same notation? If so, please make it consistence, otherwise, please use different form to avoid misunderstanding.

**Authors reply:** Equation 2 and equation 4 both use the same notation $\theta\_v$ (\theta_\mathrm{v}). Equation 3 does not include a θ. In the text at l91 however we mistakenly used a cursive 'v' (\theta_v).

**Comment 5:** For the schematic diagram in Figure 1, I suggest the authors could add some symbolic simulation obstacles in simulation section. I guess it could be the latter part in Figure 1b?

**Authors reply:** We added some symbolic obstacles to the figure.

**Comment 6:** For section 2.3.2, could you please explain in more detail about the numerical settings for lateral boundary condition, especially the inflow-outflow condition? Does it mean that the upper part of the simulation domain is used to generate flow turbulence and set as periodic, then feed into the latter part? Also, what is a "run-up" region in figure 1c?

**Authors reply:** When using inflow/outflow boundary conditions, the y direction is periodic so is still solved using FFT, but x and z are solved using cyclic reduction. The inflow boundary condition for pressure is Neumann and the outflow is Dirichlet. The bottom and top are both Neumann boundary conditions. We have clarified this in section '2.2 Method of Solution'.

The driver simulation typically uses periodic boundary conditions and is run beforehand. The velocity and scalar fields on an outflow plane are saved. These are then loaded into the target simulation with inflow-outflow boundary conditions. The outlet boundary condition is convective, using the vertically-averaged velocity profile as the outflow velocity. The run-up region is an optional region in which the flow can develop. We have added additional information in section '2.4.2 Lateral boundaries'.

**Comment 7:** Page 21, What do you mean by stating the following sentence "All processors know about the facet properties and calculate the local wall fluxes according to the state of the local fluid cells. For the facet energy balance the processor average has to be determined."? Could you please give more details for clearer understanding? Does it mean the data will be summarized in one processor via MPI and then averaged or other operation and send to other processors?

**Authors reply:** We agree that this was a confusing passage. The surface energy balance calculations are performed on a single process, and the heat flux data from the facets needs to be gathered from all the other processes. Once the surface energy balance has been calculated, the new facet temperatures are distributed to all other processes so the wall-functions can be updated. We have clarified this in section '3.7 Integration into LES'.

**Comment 8:** Refer to the previous question, does this data communication need to be done every time iteration? If so, how much time does it cost, maybe a rough comparison between data communication and computation in each time step?

**Authors reply:** Yes, the summing of the heat fluxes over the facets and their time integration is currently done at every LES time-step. These two steps could potentially be separated where the time integration is done locally and the summing is only done every energy balance time-step. However, the impact on performance will be minimal, since it is two calls to MPI_GATHER(..,MPI_SUM,..) of maximum a few MB of data, followed by two broadcasts. We added a statement that the computations related to the facets are orders of magnitudes lower than those related to the fluid cells in section '3.7 Integration into LES'.

**Comment 9:** In table 1, the grid size for main simulation is 450X400X100, and the domain size is 900X800X200. I do not think that there are only 2 grid points in x, y and z direction. So, I think the grid size should be grid number or grid points. Please revise accordingly.

**Authors reply:** Added a row 'cell size [m]:  2x2x2' so it is clearer that 450x400x100 is the dimensions in number of grid points and 900x800x200 is the dimensions in meters.

**Comment 10:** In figure 7, could you please indicate clearly about the position R?

**Authors reply:** We added a detailed view of the measurement points in the intersection to Figure 7. The measurement point R was further renamed to P1.

**Comment 11:** The test case of wind tunnel experiment results of the DAPPLE project should be briefly introduced in Section 4.2. Or relevant materials should be provided in supplements.

**Authors reply:** The sections '4.1 Validation' and '4.1.1. Simulation set-up' were revised and expanded to now include a more detailed description of the study case set-up (urban geometry etc.) and wind-tunnel experiment, including their methods of data acquisition (Carpentieri et al., 2009, 2010, 2012). Section '4.1.2 Results' now also includes quantitative

comparison between simulations and measurements using several validation metrics, with the relevant wind-tunnel data available in the supplement.

**Comment 12:** What is the test case in Figure 11 (surface energy balance test case)? This should be carefully described in the paper.

**Authors reply:** We describe now that we use a case where we try to maintain parameters describing the building morphology between the two-dimensional MTEB model and the three-dimensional uDALES. MTEB uses a street canyon representation for the built environment and e.g. the building width in the MTEB should correspond to the building area in uDALES. Added to '4.2 Verification of surface energy balance'.

**Comment 13:** In section 4.3, have you validated this case with on-site measurements? I think it will be more convincing if the authors can validate their codes with real time measurement.

**Authors reply:** Unfortunately, there are no measurements available. We have to leave the comparison with an instrumented green roof to a future study.

**Comment 14:** In section 4, have you done grid sensitivity test on these case studies? As I understand, the filter in LES simulation in this paper is implicit and based on grid size. So, the grid size should be also important for the simulation quality. Perhaps, the authors can also add turbulent kinetic energy analysis in these test cases analysis.

**Authors reply:** For the DAPPLE validation simulation, we analysed the subgrid fluxes and they do not exceed 4.5% of the total flux anywhere except for in the lowermost cells, indicating that the resolution is sufficient to resolve the majority of the energetic turbulent scales of the urban flow field. As always there is a computational cost associated with finer resolutions and we found 2m to be adequate for the problem. We now mention that the subgrid-scale fluxes are small in '4.1.1 Simulation set-up'

**Comment 15:** How is the grid arrangement of this software? Does it only allow hexagon grid? Does the grid need to be uniform in x, y and z direction?

**Authors reply:** uDALES is based on a Cartesian Arakawa C-grid (Arakawa and Lamb, 1977). The grid can be stretched in the z-direction when using periodic boundary conditions, and in the x- and z-directions when using inflow-outflow boundary conditions. We now mention this more clearly in section '2.2 Method of Solution' and section '2.3 Immersed boundary method'

My technical comments are:

**Comment 16:** Page 2, In the following sentence, I think "transition" should be "transit". "The increased likelihood of extreme weather events due to climate change (IPCC, 2014) and the need to transition to a less".

**Authors reply:** After consulting the Oxford Dictionary we believe 'transition' is correct:

'**transition,** verb: to change or to make something change from one state or condition to another'

whereas:

'**transit,** verb: (something) to pass across or through an area'

**Comment 17:** Page 3, please check the citation format in the text "require turbulence parameterisations for the full range of active scales in the flow field Blocken (2015)." Please also do this for the whole paper and revise accordingly.

**Authors reply:** We checked the entire paper for citations which were not enclosed in parentheses, but should be; and vice versa.

**Comment 18:** Page 3, please check the following sentence and revise "Popular codes for RANS are any of the commercial CFD packages (Fluent, ANSYS CFX, COMSOL etc), although there are also open source alternatives (e.g. OpenFoam)."

**Authors reply:** We rephrased the sentence, it now reads: 'There are many RANS codes available, including commercial CFD packages (Fluent, ANSYS CFX, COMSOL etc) and open source alternatives such as e.g. OpenFoam.'

**Comment 19:** Page 3, I think "large turbulent scales" rather than "bulk of the turbulent scales" in the following sentence "Large-Eddy simulation (LES) tools explicitly resolve the bulk of the turbulent scales in the flow,"

**Authors reply:** We rephrased the sentence, it now reads: 'Large-Eddy simulation (LES) tools explicitly resolve the large turbulent scales of the flow, containing the majority of the turbulent energy, and are therefore less reliant on turbulence models than RANS.'